# Parsimony or Capability? Decomposition Delivers Both in Long-term Time Series Forecasting

**Jinliang Deng**[1,2]    **Feiyang Ye**[3]    **Du Yin**[4]    **Xuan Song**[6,2*]    **Ivor Tsang**[5]    **Hui Xiong**[7,8*]

[1] Hong Kong Generative AI Research and Development Center
[2] Research Institute of Trustworthy Autonomous Systems,
Southern University of Science and Technology (SUSTech),
[3] University of Technology Sydney,    [4] University of New South Wales,
[5] CFAR and IHPC, Agency for Science, Technology and Research, Singapore,
[6] School of Artificial Intelligence, Jilin University,
[7] AI Thrust, The Hong Kong University of Science and Technology (Guangzhou),
[8] CSE, The Hong Kong University of Science and Technology
{dengjinliang, xionghui}@ust.hk,  feiyang.ye.uts@gmail.com
du.yin@unsw.edu.au,  songxuan@jlu.edu.cn,  Ivor_Tsang@cfar.a-star.edu.sg

## Abstract

Long-term time series forecasting (LTSF) represents a critical frontier in time series analysis, characterized by extensive input sequences, as opposed to the shorter spans typical of traditional approaches. While longer sequences inherently offer richer information for enhanced predictive precision, prevailing studies often respond by escalating model complexity. These intricate models can inflate into millions of parameters, resulting in prohibitive parameter scales. Our study demonstrates, through both analytical and empirical evidence, that decomposition is key to containing excessive model inflation while achieving uniformly superior and robust results across various datasets. Remarkably, by tailoring decomposition to the intrinsic dynamics of time series data, our proposed model outperforms existing benchmarks, using over 99% fewer parameters than the majority of competing methods. Through this work, we aim to unleash the power of a restricted set of parameters by capitalizing on domain characteristics—a timely reminder that in the realm of LTSF, bigger is not invariably better. The code is available at https://github.com/JLDeng/SSCNN.

## 1 Introduction

Time series forecasting is a cornerstone in the fields of data mining, machine learning, and statistics, with wide-ranging applications in finance, meteorology, city management, telecommunications, and beyond (Jiang et al., 2021; Han et al., 2023a; Zhang et al., 2020; Wu et al., 2020, 2019; Cui et al., 2023; Zhang et al., 2017; Liang et al., 2018; Zhu et al., 2024; Fan et al., 2022, 2023). Traditional univariate time series models, such as Auto-Regressive Integrated Moving Average (ARIMA) and Exponential Smoothing, fail to capture the intricate complexities present in open, dynamic systems. Fortunately, the advent of deep learning has marked a significant shift in this domain. Recently, the utilization of the Transformer (Vaswani et al., 2017) model has revolutionized time series forecasting, setting new benchmarks in the accuracy of forecasting models due to its capability of depicting intricate pairwise dependencies and extracting multi-level representations from sequences.

Inspired by the extraordinary power exhibited by large language models (LLMs), expanding model scale has increasingly become the dominant direction in the pursuit of improvement for time series

---

*Corresponding author.

38th Conference on Neural Information Processing Systems (NeurIPS 2024).

forecasting. Currently, the majority of advanced models require millions of parameters (Nie et al., 2023; Zhang & Yan, 2023; Wu et al., 2023). With the recent introduction of pre-trained large language models, such as LLaMa and GPT-2, the parameter scale has inflated to billions (Jin et al., 2024; Cao et al., 2024; Jia et al., 2024a; Zhou et al., 2023; Gruver et al., 2024). Despite the significant increase in the number of parameters to levels comparable to foundational models for language and image, the efficacy of these models has seen only marginal improvements. In particular, these large models have shown up to only a 30% improvement in MSE and MAE, yet at the cost of 100 to 1000 times more parameters compared to a simple linear model across representative tasks (Nie et al., 2023; Cao et al., 2024; Jin et al., 2024). Additionally, we have observed a convergence in the abilities demonstrated by the models since the advent of PatchTST (Nie et al., 2023), with recent advancements achieving only incremental improvements.

These evidences indicate that a large model may not necessarily be a prerequisite for the future of time series forecasting, motivating us to explore the opposite trend—minimizing the number of parameters. Before introducing our design, we reflect on why existing methods struggle to maintain their optimal effectiveness with a reduced number of parameters. Existing methods predominantly adopt data patching over either the temporal or spatial dimensions (Zhou et al., 2021; Wu et al., 2021; Zhou et al., 2022; Nie et al., 2023; Liu et al., 2024a; Zhang & Yan, 2023), which, in conjunction with the attention mechanism, allows them to capture complex temporal and spatial dependencies. However, a significant drawback of data patching is the elimination of temporal (or spatial) identities along with the destruction of temporal (or spatial) correlations, resulting in the potential loss of complex temporal (or spatial) information. To counteract this undesirable information loss, these methods establish a high-dimensional latent space to accommodate the encodings of the temporal and spatial identities in addition to the embeddings of the real-time observations. As a result, the dimensionality of the latent space typically scales with the number of identities to be encoded, inevitably leading to the exponential inflation of parameter scale. Moreover, the expansion of model size makes the models prone to overfitting, a common challenge in time series tasks where data is often limited.

To achieve a capable yet parsimonious model, it is fundamentally paramount to reinvent the paradigm to maintain and harness the spatial and temporal regularities, eliminating unnecessary and redundant parameters for encoding them into the latent space. Recent studies (Deng et al., 2021, 2024; Wang et al., 2024) showcase the potential of feature decomposition in attaining improved efficacy with limited parameters. While remarkable progress has been made, these methods struggle with long-term forecasting, especially for datasets exhibiting intricate temporal and spatial correlations, such as the Traffic and Electricity datasets (Deng et al., 2024). Moreover, the analytical aspect of decomposition along with its relation to patching is under-explored, hindering further advancements in this line of research.

In response to these limitations, we propose a Selective Structured Components-based Neural Network (SSCNN). For the first time, we address the analytical gap in feature decomposition, providing insights into its rationale for capability and parsimony compared to patching. In addition, SSCNN enhances plain feature decomposition with a selection mechanism, enabling the model to distinguish fine-grained dependencies across individual time steps, which is crucial for improving the accuracy of the decomposed structured components and, ultimately, the overall prediction accuracy. SSCNN has been benchmarked against state-of-the-art (SOTA) methods, demonstrating consistent improvements ranging from 2% to 10% in efficacy while using 99% fewer parameters than SOTA LTSF methods, including PatchTST (Nie et al., 2023) and iTransformer (Liu et al., 2024a). Remarkably, it uses 87% fewer parameters than DLinear (Zeng et al., 2023) when tasked with extensive long-term forecasting. Our contributions can be summarized as follows:

1. We introduce SSCNN, a decomposition-based model innovatively enhanced with a selection mechanism. This model is specifically designed to adeptly capture complex regularities in data while maintaining a minimal parameter scale.

2. We conduct an in-depth comparison between decomposition and patching, examining both capability and parsimony.

3. We carry out comprehensive experiments to demonstrate SSCNN's superior performance across various dimensions. These extensive evaluations not only prove its effectiveness but also highlight its versatility in handling diverse time series forecasting scenarios.

## 2 Related Work

The field of time series forecasting or spatial-temporal prediction has traditionally leveraged multi-layer perceptrons (MLPs) (Zhang et al., 2017), recurrent neural networks (RNNs) (BAI et al., 2020; Zhao et al., 2019; Jiang et al., 2023; Jia et al., 2024b), graph convolution networks (GCNs) (Yu et al., 2018), and temporal convolution networks (TCNs) (Bai et al., 2018). The recent development of ST-Norm (Deng et al., 2021), STID (Shao et al., 2022a) and STAEformer (Liu et al., 2023) shows promise in enhancing model capabilities to distinguish spatial and temporal features more effectively. Motivated by the success of self-supervised learning and pre-training in natural language processing (NLP) and computer vision (CV), these two techniques are also gaining attention and application in this field (Guo et al., 2021; Shao et al., 2022b).

Over the last few years, the focus has shifted to long-term sequence forecasting (LTSF). The majority of studies have concentrated on adapting the Transformer (Vaswani et al., 2017), successful in NLP (Devlin et al., 2018) and CV (Khan et al., 2022), for LTSF tasks. Pioneering works like LogTrans (Li et al., 2019) addressed the computational challenges of long sequences through sparse attention mechanisms. Subsequent developments, such as Informer (Zhou et al., 2021), Autoformer (Wu et al., 2021), and Fedformer (Zhou et al., 2022), introduced innovative approaches to improve predictive accuracy with temporal feature characterization, autocorrelation-based series similarities, and frequency domain conversions, respectively. Other notable contributions include the Non-stationary Transformer (Liu et al., 2022) and Triformer (Cirstea et al., 2022). Furthermore, diverse normalization techniques have been developed to mitigate the distribution shift present in time series data (Liu et al., 2024b; Kim et al., 2021). Probabilistic forecasting (Kollovieh et al., 2024) and irregular time series forecasting (Chen et al., 2024; Ansari et al., 2023) are two growing subfields receiving increasing attention.

A significant shift in LTSF research occurred with DLinear (Zeng et al., 2023), an embarrassingly simple linear model. DLinear highlighted the limitation of Transformers in capturing the unique ordering information of time series data (Zeng et al., 2023). To overcome this limitation, recent methods like PatchTST (Nie et al., 2023), TimesNet (Wu et al., 2023), Crossformer (Zhang & Yan, 2023), and iTransformer (Liu et al., 2024a) blend the global dependency capabilities of Transformers with the local order modeling strengths of MLPs. However, the success of these methods is achieved at the cost of an enormous number of parameters.

A handful of emerging studies have sought to reduce parameter usage in pursuit of a parsimonious model (Lin et al., 2024; Xu et al., 2024; Wang et al., 2024; Deng et al., 2024). The techniques they employed to remove redundancies fall into three categories: downsampling (Lin et al., 2024), decomposition (Deng et al., 2024; Wang et al., 2024), and Fourier transform (Xu et al., 2024; Yi et al., 2024). Despite efficient parameter usage, these methods often compromise accuracy for specific datasets, especially those presenting complex yet predictable patterns, such as Traffic (Deng et al., 2024; Xu et al., 2024). *Our study, as far as we know, is the first to realize a parsimonious model without any sacrifice of capability.*

## 3 Selective Structured Components-based Neural Network

In multivariate time series forecasting, given historical observations $\mathbf{X} = \{\mathbf{x}_1, \cdots, \mathbf{x}_N\} \in \mathbb{R}^{N \times T_{\text{in}}}$ with $N$ variates and $T_{\text{in}}$ time steps, we predict the future $T_{\text{out}}$ time steps $\hat{\mathbf{X}} \in \mathbb{R}^{N \times T_{\text{out}}}$. The input data goes through the processing, visualized in Fig. 1, for predicting the unknown, future data. Over the course of prediction, a sequence of intermediate representations are yielded. Essentially, SSCNN is structured into two distinct branches: the top branch illustrates the inference process used to derive the components accompanied by the residuals, while the bottom branch depicts the extrapolation process, forecasting the potential evolution of these components. The components and residuals obtained are combined into a wide vector, which is then input into a polynomial regression layer to capture their complex interrelations. In the following sections, we detail the inference and extrapolation processes for these components, respectively.

### 3.1 Temporal Component

We invent temporal attention-based normalization (T-AttnNorm) to decompose the temporal components, consisting of the long-term component, the seasonal component, and the short-term component,

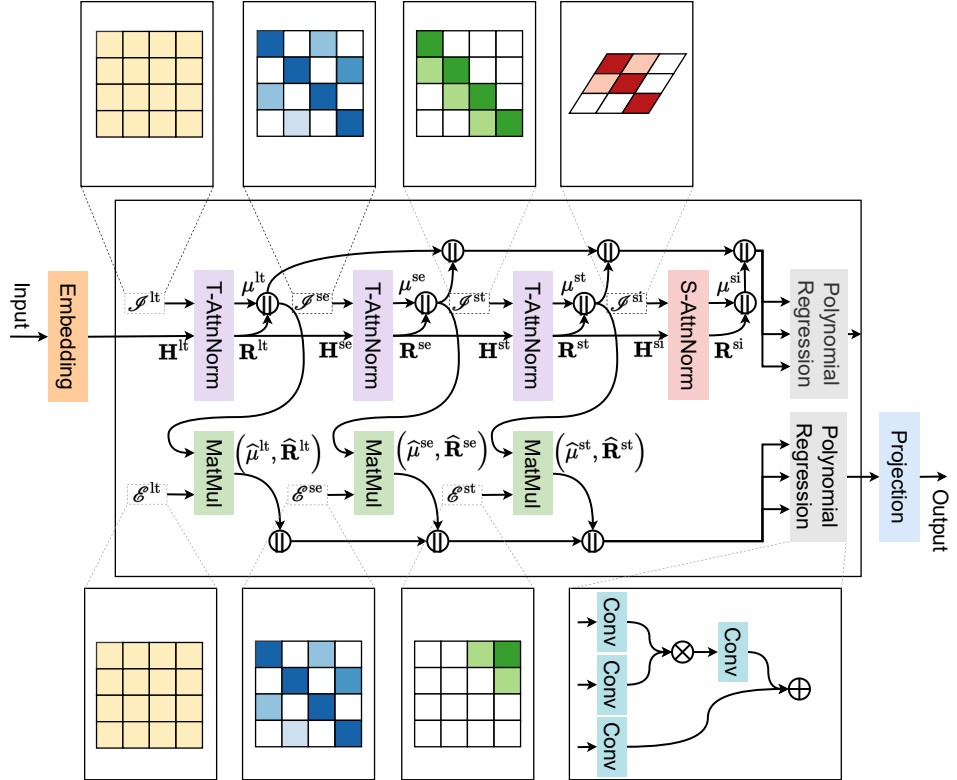

Figure 1: An overview of the SSCNN. The grids are used to exemplify the selection maps $\mathcal{I}^*$ and $\mathcal{E}^*$ as defined in the main text, with $T_{\text{in}}$, $T_{\text{out}}$ and $N$ instantiated as 4, 4 and 3, respectively.

in a sequential manner. The inference for each of these three components is applied to the representation of each series individually along the temporal dimension, with a selection/attention map characterizing the dynamics of the component of interest. Upon inference, our model disentangles the derived structured component from the data, resulting in a residual term summarizing other remaining components. The initial representation of the time series, the derived structured component, and the residual term are denoted as $\mathbf{H}^*, \mu^*, \mathbf{R}^* \in \mathbb{R}^{N \times T_{\text{in}} \times d}$, respectively. The selection map is denoted as $\mathcal{I}^* \in \mathbb{R}^{T_{\text{in}} \times T_{\text{in}}}$. T-AttnNorm is formulated as follows:

$$\mu_i^*, \mathbf{R}_i^* = \text{T-AttnNorm}(\mathbf{H}_i^*; \mathcal{I}^*), \tag{1}$$

where $\mu_i^* = \mathcal{I}^* \mathbf{H}_i^*$, $\sigma_i^{*2} = \mathcal{I}^* \mathbf{H}_i^{*2} - \mu_i^{*2} + \epsilon$, $\mathbf{R}_i^* = \frac{\mathbf{H}_i^* - \mu_i^*}{\sigma_i^*}$. To ensure the unbiasedness of $\mu_i^*$ and $\sigma_i^*$, the sum of each row of $\mathcal{I}^*$ is constrained to 1. The distinction between the three components is the realization of $\mathcal{I}^*$. The residual term resulting from the current block is taken as the input for the subsequent block, e.g., $\mathbf{H}^{\text{se}} = \mathbf{R}^{\text{lt}}$.

Then, the component series and the residual series are respectively extrapolated to forward horizons by a mapping parameterized with $\mathcal{E}^* \in \mathbb{R}^{T_{\text{out}} \times T_{\text{in}}}$:

$$\hat{\mu}_i^*, \hat{\mathbf{R}}_i^* = \text{Extrapolate}(\mu_i^*, \mathbf{R}_i^*; \mathcal{E}^*), \tag{2}$$

where $\hat{\mu}_i^* = \mathcal{E}^* \mu_i^*$, $\hat{\mathbf{R}}_i^* = \mathcal{E}^* \mathbf{R}_i^*$, with $\hat{\mu}_i^*, \hat{\mathbf{R}}_i^* \in \mathbb{R}^{T_{\text{out}} \times d}$, leading to $\hat{\mathbf{R}}^*, \hat{\mu}^* \in \mathbb{R}^{N \times T_{\text{out}} \times d}$. Similar to $\mathcal{I}^*$, $\mathcal{E}^*$ is also defined as a matrix with the sum of each row equal to 1. Next, we introduce how to realize $\mathcal{I}^*$ and $\mathcal{E}^*$ to extract and simulate the dynamics of the considered components, respectively.

**Long-Term Component**  The long-term component aims to characterize the trend patterns of the time series data. To acquire an estimation of the long-term component with less bias, we aggregate the samples collected across multiple seasons, eliminating the seasonal and short-term impacts that impose only local effects. The realizations of the inference and extrapolation matrices for the

long-term component are given by:

$$\mathcal{I}^{\text{lt}}(i, j) = \frac{1}{T_{\text{in}}}, \ \mathcal{E}^{\text{lt}}(i, j) = \frac{1}{T_{\text{in}}}. \tag{3}$$

The attention mechanism is excluded from the long-term component as it does not improve forecasting accuracy in our datasets. The attention mechanism is beneficial when a component's distribution changes significantly over time, helping to reduce estimation bias. However, for our long-term component, the distribution remains stable throughout the input period. In such cases, the attention mechanism adds no value, making it redundant and unnecessary.

**Seasonal Component**   The seasonal component is created to model the seasonal fluctuation. Inference of the seasonal component operates under the assumption of a consistent cycle duration to streamline the detection of seasonal trends. We introduce $c$ to indicate the length of a cycle, $\tau_{\text{in}}$ to signify the maximal count of cycles encompassed by the input sequence, i.e., $\tau_{\text{in}} c \leq T_{\text{in}}$, and $\tau_{\text{out}}$ to denote the minimal number of cycles covering the output sequence, i.e., $\tau_{\text{out}} c \geq T_{\text{out}}$. To simplify the notation, we assume that $T_{\text{in}}$ is a multiple of $c$, i.e., $T_{\text{in}} = \tau_{\text{in}} \cdot c$.

To acquire unbiased and precise seasonal components, we introduce a parameter matrix $\mathbf{W}^{\text{se}} \in \mathbb{R}^{\tau_{\text{in}} \times \tau_{\text{in}}}$, allocating distinct weights to each pair of periods to represent the inter-cycle correlations. The matrix undergoes row-wise normalization via a softmax operation to satisfy the constraint on the sum of 1. The selection map for inferring the seasonal component is defined as:

$$\mathcal{I}^{\text{se}}(i, j) = \begin{cases} \frac{\exp(\mathbf{W}^{\text{se}}_{u,v})}{\sum_{k=0}^{\tau_{\text{in}}-1} \exp(\mathbf{W}^{\text{se}}_{u,k})} & u = \lfloor \frac{i}{c} \rfloor, v = \lfloor \frac{j}{c} \rfloor, i - j \equiv 0 \pmod{c} \\ 0 & \text{Otherwise} \end{cases}, \tag{4}$$

where $\lfloor \cdot \rfloor$ denotes the floor function.

When dealing with extrapolation, we define $\hat{\mathbf{W}}^{\text{se}} \in \mathbb{R}^{\tau_{\text{out}} \times \tau_{\text{in}}}$ as the parameter matrix capturing the correlations between each pair of cycles encompassed by the input and output sequences, respectively. The selection map for extrapolating the seasonal component is written as follows:

$$\mathcal{E}^{\text{se}}(i, j) = \begin{cases} \frac{\exp(\hat{\mathbf{W}}^{\text{se}}_{u,v})}{\sum_{k=0}^{\tau_{\text{in}}-1} \exp(\hat{\mathbf{W}}^{\text{se}}_{u,k})} & u = \lfloor \frac{i}{c} \rfloor, v = \lfloor \frac{j}{c} \rfloor, i - j \equiv 0 \pmod{c} \\ 0 & \text{Otherwise} \end{cases}. \tag{5}$$

**Short-Term Component**   The short-term component discerns irregularities and ephemeral phenomena unaccounted for by the seasonal and long-term components. In contrast to the long-term component, it necessitates only a limited window size, $\delta$, encapsulating recent observations with immediate relevance. Moreover, these observations exhibit varying degrees of correlation depending on the associated lag. Therefore, the inference of the short-term component involves a parameter vector $\mathbf{w}^{\text{st}} \in \mathbb{R}^{\delta}$, and is expressed mathematically as:

$$\mathcal{I}^{\text{st}}(i, j) = \begin{cases} \frac{\exp(\mathbf{w}^{\text{st}}_i)}{\sum_{i=0}^{\delta-1} \exp(\mathbf{w}^{\text{st}}_i)} & (i - j >= 0) \wedge (i - j < \delta) \\ 0 & \text{Otherwise} \end{cases}. \tag{6}$$

The extrapolation of the short-term component bifurcates based on the targeted horizon. Immediate horizons retain correlations with preceding estimations of the short-term component, prompting regression-based forecasting with a parameter vector $\hat{\mathbf{w}}^{\text{st}} \in \mathbb{R}^{\delta \times \delta}$. Conversely, as the horizon extends ahead, it accumulates compounded uncertainties, decreasing the predictability. Herein, we opt for zero-padding to eliminate unnecessary parameters. The entire extrapolation is formalized as follows:

$$\mathcal{E}^{\text{st}}(i, j) = \begin{cases} \frac{\exp(\hat{\mathbf{w}}^{\text{st}}_{i,j})}{\sum_{k=0}^{\delta-1} \exp(\hat{\mathbf{w}}^{\text{st}}_{i,k})} \cdot & (i < \delta) \wedge (j > T_{\text{in}} - \delta - 1) \\ 0 & \text{Otherwise} \end{cases}. \tag{7}$$

## 3.2   Spatial Component

The spatial component refers to the component that is temporally irregular, i.e., cannot be captured by the aforementioned three temporal components, but spatially regular, i.e., showing consistent behavior across a group of residual series. The inference of the spatial component, referred to

as spatial attention-based normalization (S-AttnNorm), shares a similar formula representation as the temporal component, except that it is applied to each frame independently along the spatial dimension:

$$\mu^*_{:,t}, \mathbf{R}^*_{:,t} = \text{S-AttnNorm}(\mathbf{H}^*_{:,t}; \mathcal{I}^*), \tag{8}$$

where $\mu^*_{:,t} = \mathcal{I}^* \mathbf{H}^*_{:,t}$, $\sigma^{*}_{:,t}{}^2 = \mathcal{I}^* \mathbf{H}^*_{:,t}{}^2 - \mu^*_{:,t}{}^2 + \epsilon$, $\mathbf{R}^*_{:,t} = \frac{\mathbf{H}^*_{:,t} - \mu^*_{:,t}}{\sigma^*_{:,t}}$. We acquire the similarities among the series in terms of the spatial component by applying correlation to the residuals post controlling the long-term, seasonal, and short-term components. Given that each series is represented by a $T_{\text{in}} \times d$ matrix in the hidden layer, we vectorize them before performing the measurement. This results in a similarity matrix $\mathcal{I}^{\text{si}} \in \mathbb{R}^{N \times N}$, with each entry $\mathcal{I}^{\text{si}}(i, j)$ representing the conditional correlation between the $i^{\text{th}}$ series and $j^{\text{th}}$ series:

$$\mathcal{I}^{\text{si}}(i, j) = \frac{\exp(\text{vec}(\mathbf{H}^{\text{si}}_i)^\top \text{vec}(\mathbf{H}^{\text{si}}_j))}{\sum_{k=1}^{N} \exp(\text{vec}(\mathbf{H}^{\text{si}}_i)^\top \text{vec}(\mathbf{H}^{\text{si}}_k))}. \tag{9}$$

Considering the erratic and unpredictable nature of the spatial component along time, we simply realize component extrapolation with zero-padding: $\hat{\mu} = \mathbf{0}$ and $\hat{\mathbf{R}} = \mathbf{0}$.

### 3.3 Component Fusion

The Polynomial Regression layer is inspired by the work of Deng et al. (2024), where we extend the original module to include both additive and multiplicative relations. This extension allows us to model more complex interactions between the decomposed components.

$$\mathbf{H}_i = \text{Conv}_{k=1}\left(\text{Conv}_k(\mathbf{S}_i) \otimes \text{Conv}_k(\mathbf{S}_i)\right) + \text{Conv}_k(\mathbf{S}_i), \tag{10}$$

where

$$\mathbf{S}_i = \left[\mathbf{R}^{\text{lt}}_i, \mu^{\text{lt}}_i, \mathbf{R}^{\text{se}}_i, \mu^{\text{se}}_i, \mathbf{R}^{\text{st}}_i, \mu^{\text{st}}_i, \mathbf{R}^{\text{si}}_i, \mu^{\text{si}}_i\right].$$

The resulting $\mathbf{H}_i$ is fed to the next layer as $\mathbf{H}^{\text{lt}}_i$ for further manipulation.

## 4 Comparison Between Decomposition and Patching

**Capability Analysis** We first demonstrate that the representation space accommodating only a plain single-step observation is ill-structured due to the entanglement of diverse component signals. Then, we elucidate how patching and decomposition adjust the structure of the representation space, facilitating the capture of faithful and reliable relations among the samples. The detailed derivations for this section are provided in Appendix A.

For simplicity, we assume that time series $x$ is driven by two distinct components $a$ and $b$, but the analysis can be trivially extended to cases with multiple components. Consider a triplet $(x_{t_1}, x_{t_2}, x_{t_3})$ collected at three time steps $t_1$, $t_2$, and $t_3$, respectively, where $x_{t_i} = a_{t_i} + b_{t_i}$ for $i = 1, 2, 3$, subject to $a_{t_1} = a_{t_2}$, $b_{t_2} = b_{t_3}$, and $a_{t_1} \neq a_{t_3}$. We expect that $x_{t_1}$ should be closer to $x_{t_2}$ than $x_{t_3}$, given $\|a_{t_1} - a_{t_2}\| < \|a_{t_1} - a_{t_3}\|$ along with $\|b_{t_1} - b_{t_2}\| = \|b_{t_1} - b_{t_3}\|$. However, this relationship may not necessarily hold with step-wise observations, i.e., there exist specified triplets $(\hat{x}_{t_1}, \hat{x}_{t_2}, \hat{x}_{t_3})$ such that $\|\hat{x}_{t_1} - \hat{x}_{t_2}\| > \|\hat{x}_{t_1} - \hat{x}_{t_3}\|$, due to the entanglement of $a$ and $b$.

Patching can rectify sample relations by creating orthogonality between distinct components, resulting in a well-structured representation space. By incrementally augmenting the time series representation with preceding observations, i.e., $[x_{t-p}, \cdots, x_t]$, the $a$-component ($[a_{t-p}, \cdots, a_t]$) and the $b$-component ($[b_{t-p}, \cdots, b_t]$) become increasingly orthogonal, reducing negative interference caused by their interactions. It is worth noting that while patching is widely acknowledged for enriching the semantic information of step-wise data (Nie et al., 2023), no formal explanation from the perspective of disentanglement has been offered yet.

Distinct from patching, which implicitly attenuates the interactions among distinct components, decomposition can explicitly avert any interactions. This is done by first recovering and subsequently distributing different components into disjoint sets of dimensions, completely isolating them in independent subspaces. Specifically, this involves finding a decomposition mapping $D : a_{t_i} + b_{t_i} \rightarrow$

$(a_{t_i}, b_{t_i})^\top$ such that $\|D(x_{t_1}) - D(x_{t_2})\| \leq \|D(x_{t_1}) - D(x_{t_3})\|$. The effectiveness of decomposition depends on how accurately the model can recover $a$ and $b$. Thus, it is paramount to ensure that no irrelevant information is included in estimating the component of interest, suggesting the necessity of using a selection mechanism. The above analysis is based on deterministic data, but similar results can be obtained if the data possesses stochasticity, which is the case for real-world applications. See detailed discussion in Appendix A.

**Parsimony Analysis** To justify that SSCNN is fundamentally more parameter-efficient than patching-based methods, we analyze how the number of required parameters scales with $T_{\text{in}}$, $T_{\text{out}}$, and $d_{\text{patch}}/d_{\text{step}}$, where $d_{\text{patch}}$ and $d_{\text{step}}$ represent the dimensionalities of the representation spaces for patching-based and decomposition-based methods, respectively.

Patching-based methods establish intense connections from backward variables to forward variables via multiple layers of patch-wise fully-connected networks, typically necessitating $\mathcal{O}(d_{\text{patch}}T_{\text{in}}T_{\text{out}} + d_{\text{patch}}^2)$ parameters (Nie et al., 2023). Empirically, the optimal performance of these models is achieved when $d_{\text{patch}}$ is adjusted within the range of 64 and 256, resulting in an explosion in the total number of parameters, especially for large $T_{\text{in}}$ and $T_{\text{out}}$.

In the case of SSCNN, the portion scaling with $T_{\text{in}}$, $T_{\text{out}}$, and $d_{\text{step}}$ contains $\mathcal{O}(d_{\text{step}}^2 + \frac{1}{c^2}T_{\text{in}}T_{\text{out}} + \frac{1}{c^2}T_{\text{in}}^2 + d_{\text{step}}T_{\text{out}})$ parameters. These come from three sources: the convolution operators in the polynomial regression module, the attention map responsible for characterizing seasonal patterns, and the predictor. In contrast to $d_{\text{patch}}$, $d_{\text{step}}$ can be assigned a small number, e.g., 8, which is sufficient to prompt the model to exhibit its full potential, making $d_{\text{step}}^2$ orders of magnitude smaller than $d_{\text{patch}}^2$. Additionally, SSCNN significantly reduces the number of connections scaling with $T_{\text{in}}$ and $T_{\text{out}}$ from $d_{\text{patch}}T_{\text{in}}T_{\text{out}}$ to $\frac{1}{c^2}T_{\text{in}}T_{\text{out}} + \frac{1}{c^2}T_{\text{in}}^2 + d_{\text{step}}T_{\text{out}}$, thanks to the capability of decomposition in eliminating redundant correlations across variables, as further analyzed in Appendix D. The presence of the scaling factor $\frac{1}{c^2}$ makes SSCNN even more parameter-efficient than DLinear (Zeng et al., 2023) for large $T_{\text{in}}$ and $T_{\text{out}}$, when the sum of terms related to $T_{\text{in}}$ and $T_{\text{out}}$ dominates $\mathcal{O}(d_{\text{step}}^2)$. The actual results are presented in Sec. 5.2.

# 5 Evaluations

In this section, we conduct comprehensive experiments across standard datasets to substantiate from various perspectives that SSCNN achieves SOTA performance with a minimal parameter count.

## 5.1 Experiment Setting

**Datasets** We evaluate the performance of our proposed SSCNN on seven popular datasets with diverse regularities, including Weather, Traffic, Electricity, and four ETT datasets (ETTh1, ETTh2, ETTm1, ETTm2). Among these seven datasets, Traffic and Electricity consistently show more regular patterns over time, while the rest contain more volatile data. Detailed dataset descriptions and data processing are provided in Appendix B.

**Evaluation Metrics** In line with established practices in LTSF (Nie et al., 2023; Wu et al., 2021), we evaluate the model performance using mean squared error (MSE) and mean absolute error (MAE).

**Baseline Models** We compare SSCNN with the state-of-the-art models, including (1) Transformer-based methods: Autoformer (Wu et al., 2021), Crossformer (Zhang & Yan, 2023), PatchTST (Nie et al., 2023) and iTransformer Liu et al. (2024a), (2) Linear-based method: DLinear Zeng et al. (2023); (3) TCN-based method: TimesNet Wu et al. (2023); (4) Decomposition-based method: TimeMixer (Wang et al., 2024), SCNN (Deng et al., 2024). For implementing state-of-the-art models (SOTAs), we adhere to the default settings as provided in the Time-Series-Library [2].

**Network Setting** All the experiments are implemented in PyTorch and conducted on a single NVIDIA 2080Ti 11GB GPU. For ECL and Traffic, SSCNN is configured with 4 layers, and the number of hidden channels $d$ is set to 8. We set $\delta$ to a value from $\{2, 4, 8, 16\}$. The kernel size for the

---

[2]https://github.com/thuml/Time-Series-Library

Table 1: Multivariate forecasting results with prediction lengths S ∈ {3, 24, 96, 192, 336}. Results are averaged from all prediction lengths. Full results are listed in Appendix C. The best result is highlighted in **bold**, and the second best is highlighted with underline.

| Models | SSCNN (Ours) | | SCNN (2024) | | iTransformer (2024) | | TimeMixer (2024) | | PatchTST (2023) | | TimesNet (2023) | | Crossformer (2023) | | DLinear (2023) | | Autoformer (2021) | |
|---|---|---|---|---|---|---|---|---|---|---|---|---|---|---|---|---|---|---|
| Metric | MSE | MAE | MSE | MAE | MSE | MAE | MSE | MAE | MSE | MAE | MSE | MAE | MSE | MAE | MSE | MAE | MSE | MAE |
| ECL | **0.118** | **0.212** | 0.128 | 0.222 | 0.121 | 0.216 | 0.123 | 0.215 | 0.125 | 0.219 | 0.163 | 0.267 | 0.124 | 0.221 | 0.141 | 0.236 | 0.191 | 0.308 |
| Traffic | **0.338** | **0.235** | 0.360 | 0.254 | 0.343 | 0.246 | 0.387 | 0.271 | 0.349 | 0.241 | 0.579 | 0.314 | 0.375 | 0.254 | 0.426 | 0.290 | 0.583 | 0.420 |
| ETTh1 | **0.330** | **0.363** | 0.339 | 0.368 | 0.359 | 0.387 | 0.348 | 0.375 | 0.335 | 0.372 | 0.399 | 0.418 | 0.349 | 0.378 | 0.368 | 0.393 | 0.452 | 0.466 |
| ETTh2 | **0.255** | **0.315** | 0.257 | **0.315** | 0.274 | 0.331 | 0.264 | 0.324 | 0.264 | 0.322 | 0.304 | 0.356 | 0.289 | 0.338 | 0.282 | 0.338 | 0.365 | 0.411 |
| ETTm1 | **0.242** | **0.300** | 0.244 | 0.302 | 0.261 | 0.315 | 0.258 | 0.316 | 0.248 | 0.305 | 0.271 | 0.318 | 0.272 | 0.318 | 0.259 | 0.312 | 0.464 | 0.445 |
| ETTm2 | **0.158** | 0.236 | **0.158** | **0.235** | 0.175 | 0.251 | 0.164 | 0.241 | 0.167 | 0.240 | 0.180 | 0.258 | 0.168 | 0.244 | 0.167 | 0.251 | 0.225 | 0.306 |
| Weather | **0.139** | **0.175** | 0.140 | 0.178 | 0.158 | 0.190 | 0.143 | 0.182 | 0.153 | 0.184 | 0.165 | 0.206 | 0.150 | 0.194 | 0.161 | 0.209 | 0.185 | 0.231 |

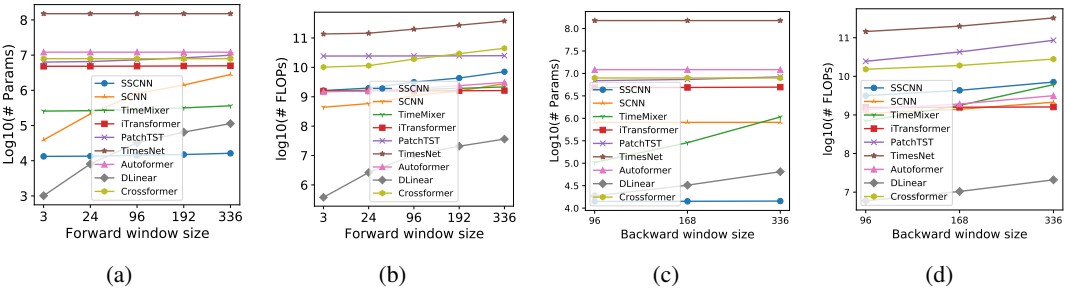

(a)        (b)        (c)        (d)

Figure 2: Examination of parameter scale and computation scale against the forward window size and the backward window size on the ECL dataset.

convolution used in polynomial regression is chosen as 2. The total number of parameters resulting from this configuration is around **25,000**. For the other five datasets showing unstable and volatile patterns, the spatial component is found to be useless due to the absence of spatial correlations, thus this component is disabled. Additionally, the number of layers is configured as 2, resulting in around **5,000** parameters. During the training phase, we employ L2 loss for model optimization. In addition, the batch size is set to 8, and an Adam optimizer is used with a learning rate of 0.0005.

## 5.2 Comparative Analysis with Baselines

**Forecasting Accuracy** As reported in Table 1, for Electricity and Traffic, SSCNN showcased remarkable proficiency by achieving the lowest MSE and MAE. Specifically, on the Electricity dataset, SSCNN outperformed iTransformer and PatchTST, two representative baselines, by 2% and 4%, respectively. On the Traffic dataset, SSCNN again surpassed them by 2% and 3%, respectively. By comparing SSCNN to SCNN, the benefit of the selection mechanism is evident, leading to approximate 8% improvement, highlighting the necessity of modeling fine-grained correlations for complex yet regular patterns. On the other five datasets, we find that the attention mechanism benefits little, resulting in comparable performance between SSCNN and SCNN. When it comes to the comparative analysis with three leading Transformer-based models, including iTransformer, PatchTST, and Crossformer, SSCNN exhibits notable improvements by 8.8%, 4.4%, and 8.3%, respectively, on average. This suggests the competitiveness of SSCNN in handling irregular dynamics.

**Parameter Scale** Parameter scale is represented by the number of parameters. This metric varies with the lookback window size and the forward window size for all models, as illustrated in Fig. 2a and Fig. 2c, respectively. Apparently, SSCNN and DLinear emerge as the top two parameter-efficient models, requiring **99%** fewer parameters than other models. Remarkably, SSCNN proves to be even more parsimonious than DLinear when tasked with extensively long-term forecasting, showing a considerable reduction of **87%** in parameter scale at the forward window size of 336.

**Computation Scale** Computation scale is measured using the number of floating point operations (FLOPs), as illustrated in Fig. 2b and Fig. 2d, respectively. SSCNN falls short in computation scale compared to iTransformer. This is because, instead of performing compression using fully connected layers, it manages and processes the entire sequence at each layer. As a result, the computational

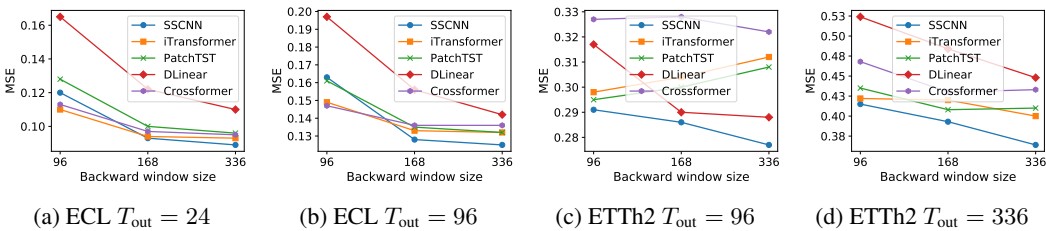

| (a) ECL $T_{\text{out}} = 24$ | (b) ECL $T_{\text{out}} = 96$ | (c) ETTh2 $T_{\text{out}} = 96$ | (d) ETTh2 $T_{\text{out}} = 336$ |

Figure 3: Impacts of backward window size.

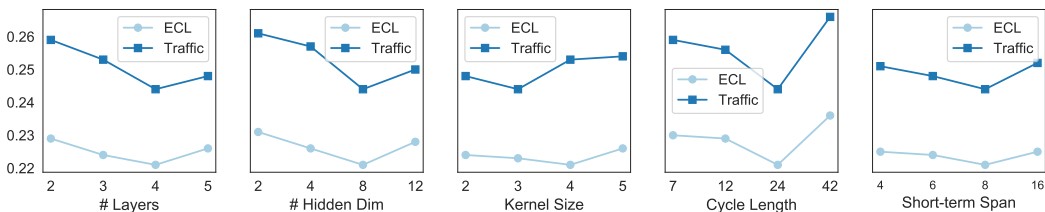

Figure 4: Sensitivity analysis of hyper-parameters on the ECL and Traffic datasets.

cost scales with the sum of the backward window and forward window sizes, raising an issue to be addressed in future work. Despite the relatively high computational cost, SSCNN has the potential to be accelerated through parallel computing.

**Sensitivity to Lookback Window Size**  As shown in Fig. 3, for ECL data, although SSCNN initially lags behind other state-of-the-art models at a window size of 96, it progressively outperforms them as more historical data is included. In contrast, iTransformer and Crossformer achieve advantageous performance with shorter input ranges but exhibit diminished gains from extended historical data. Furthermore, when handling the volatile fluctuations of ETTh2 data, SSCNN demonstrates indisputably enhanced capability with respect to the lookback window size compared to competing methods, some of which even show degraded performance due to the overfitting problem.

### 5.3 Comparative Analysis of Model Configurations

**Analysis of Hyper-parameters**  As shown in Fig. 4, the efficacy of SSCNN is impacted by adjustments to the five considered hyperparameters, showing up to a 10% difference between the best and the worst records. Focusing on the performance change against cycle length, we find that the misalignment between the configured value and the authentic value of 24 results in a significant drop in effectiveness, verifying our presumption. Additionally, the short-term span also has a non-negligible influence on the outcome.

**Ablation Study**  We create seven variants: The first six variants are used to assess the individual contributions of the four components as well as the two introduced attention maps, respectively. The last variant, represented as 'w FCN', is constructed by inserting an additional fully-connected network (FCN) between backward variables and forward variables to verify the redundancy of the FCN, which is prevalently adopted by previous works (Liu et al., 2024a; Nie et al., 2023) to capture temporal correlations, under the framework of SSCNN. It is evident from Fig. 5 that all these components

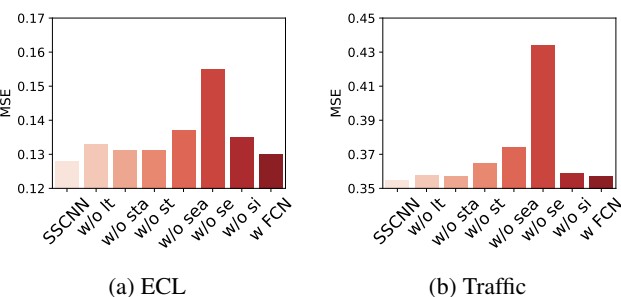

| (a) ECL | (b) Traffic |

Figure 5: Performance comparison with various component in ECL and Traffic dataset.

are vital, with the seasonal component being the most prominently important. Interestingly, our approach shows that basing dependencies on the evaluation of inter-channel conditional correlations actually enhances outcomes, contradicting the evidence delivered by Han et al. (2023b); Nie et al. (2023), where a channel-dependent strategy was seen as detrimental to model performance. Furthermore, we note that 'w FCN' does not bring any benefit, proving the sufficiency of the proposed inference and extrapolation operations for characterizing useful dynamics.

## 6    Conclusions, Limitations and Broader Impacts

In this research, we developed SSCNN, a structured component-based neural network augmented with a selection mechanism, adept at capturing complex data regularities with a reduced parameter scale. Through extensive experimentation, SSCNN has demonstrated exceptional performance and versatility, effectively handling a variety of time series forecasting scenarios and proving its efficacy.

While the proposed SSCNN model achieves satisfactory results in reducing parameter usage, it falls short in terms of computational complexity, as shown in Fig. 2b and Fig. 2d. This drawback, however, has the potential to be mitigated in future work by identifying and eliminating redundant computations with down-sampling techniques. Moreover, our future research will explore the potential for automating the process of identifying the optimal neural architecture, using these fundamental modules and operations as building blocks. This approach promises to alleviate the laborious task of manually testing various combinations in search of the optimal architecture for each new dataset encountered.

Time series forecasting can significantly benefit various sectors, including meteorology, economics, traffic management, and healthcare. However, it also poses potential negative impacts. Inaccurate forecasts can lead to significant economic or operational disruptions. Additionally, the large amounts of data required for accurate forecasting may include personal or sensitive information, raising concerns about data privacy and potential misuse.

## 7    Acknowledgment

The research was supported by Theme-based Research Scheme (T45-205/21-N) from Hong Kong RGC, Generative AI Research and Development Centre from InnoHK, and the Open Project Program of State Key Laboratory of Virtual Reality Technology and Systems, Beihang University (No.VRLAB2024A**). This work was also partially supported by the grants of National Key Research and Development Project (2021YFB1714400) of China, Jilin Provincial International Cooperation Key Laboratory for Super Smart City and Zhujiang Project (2019QN01S744). This work was supported in part by the National Key R&D Program of China (Grant No.2023YFF0725001),in part by the National Natural Science Foundation of China (Grant No.92370204), in part by the guangdong Basic and Applied Basic Research Foundation (Grant No.2023B1515120057), and in part by Guangzhou-HKUST(GZ) Joint Funding Program (Grant No.2023A03J0008), Education Bureau of Guangzhou Municipality.

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

# A  Additional Materials for Section 4

Suppose the time series $x$ is driven by two distinct vector components $a$ and $b$. Considering a triplet $(x_{t_1}, x_{t_2}, x_{t_3})$ sampled from three time steps $t_1$, $t_2$ and $t_3$, respectively, where $x_{t_i} = a_{t_i} + b_{t_i}$ for $i = 1, 2, 3$, subjecting to $a_{t_1} = a_{t_2}$, $b_{t_2} = b_{t_3}$ and $a_{t_1} \neq a_{t_3}$. To examine the relationship between $\|x_{t_1} - x_{t_2}\|$ and $\|x_{t_1} - x_{t_3}\|$, we analyze the difference between the squares of them:

$$
\begin{aligned}
\|x_{t_1} - x_{t_2}\|^2 - \|x_{t_1} - x_{t_3}\|^2 &= \|a_{t_1} + b_{t_1} - a_{t_2} - b_{t_2}\|^2 - \|a_{t_1} + b_{t_1} - a_{t_3} - b_{t_3}\|^2 \\
&= \|b_{t_1} - b_{t_2}\|^2 - \|a_{t_1} + b_{t_1} - a_{t_3} - b_{t_2}\|^2 \\
&= -\|a_{t_1} - a_{t_3}\|^2 - 2\langle a_{t_1} - a_{t_3}, b_{t_1} - b_{t_2}\rangle.
\end{aligned}
\tag{11}
$$

Therefore, the sign of the difference is determined by the relation between $a_{t_1} - a_{t_3}$ and $b_{t_1} - b_{t_3}$. If $a_{t_1} - a_{t_3}$ and $b_{t_1} - b_{t_3}$ are orthogonal, implying that their inner product is zero, then Eq. (11) can be further reduced to $-\|a_{t_1} - a_{t_3}\|^2$, resulting in a rigorously negative number, which means that $x_{t_1}$ is constantly closer to $x_{t_2}$ than $x_{t_3}$. Whereas, if they are not perpendicular to each other, which is the case for the step-wise representations of time series data, the sign can be arbitrary. To make it more accessible, we express $b_{t_1} - b_{t_3}$ as the addition of two components, i.e., $v_1 + v_2$, where $v_1 = \lambda(a_{t_1} - a_{t_3})$, representing the projection of $b_{t_1} - b_{t_3}$ on $a_{t_1} - a_{t_3}$, and $v_2 = b_{t_1} - b_{t_3} - v_1$, explaining the residual part. Substituting this expression of $b_{t_1} - b_{t_3}$ into Eq. (11), we obtain $-(1 + 2\lambda)\|a_{t_1} - a_{t_3}\|^2$. At this point, it is easy to capture that the relationship between $\|x_{t_1} - x_{t_2}\|$ and $\|x_{t_1} - x_{t_3}\|$ depends on the value taken by $\lambda$. Therefore, $x_{t_1}$ is likely to be farther to $x_{t_2}$ than $x_{t_3}$, even $x_{t_1}$ and $x_{t_2}$ share one same component.

The patching operation, i.e., $P : x_t \to (x_{t-p}, \cdots, x_t)^\top$, restructures the representation space by adjusting the distances among the samples. In particular, the difference between $\|x_{t_1} - x_{t_2}\|$ and $\|x_{t_1} - x_{t_3}\|$ is transformed into the following form:

$$
\begin{aligned}
&\|P(x_{t_1}) - P(x_{t_2})\|^2 - \|P(x_{t_1}) - P(x_{t_3})\|^2 \\
&= \|(x_{t_1-p} - x_{t_2-p}, \cdots, x_{t_1} - x_{t_2})\|^2 - \|(x_{t_1-p} - x_{t_3-p}, \cdots, x_{t_1} - x_{t_3})\|^2 \\
&= \|(a_{t_1-p} - a_{t_2-p}, \cdots, a_{t_1} - a_{t_2})\|^2 + \|(b_{t_1-p} - b_{t_2-p}, \cdots, b_{t_1} - b_{t_2})\|^2 \\
&\quad + 2\langle(a_{t_1-p} - a_{t_2-p}, \cdots, a_{t_1} - a_{t_2}), (b_{t_1-p} - b_{t_2-p}, \cdots, b_{t_1} - b_{t_2})\rangle \\
&\quad - \|(a_{t_1-p} - a_{t_3-p}, \cdots, a_{t_1} - a_{t_3})\|^2 - \|(b_{t_1-p} - b_{t_3-p}, \cdots, b_{t_1} - b_{t_3})\|^2 \\
&\quad - 2\langle(a_{t_1-p} - a_{t_3-p}, \cdots, a_{t_1} - a_{t_3}), (b_{t_1-p} - b_{t_3-p}, \cdots, b_{t_1} - b_{t_3})\rangle.
\end{aligned}
\tag{12}
$$

Given that $a$ and $b$ display independent dynamics, the correlation between $(a_{t-p}, \cdots, a_t)$ and $(b_{t-p}, \cdots, b_t)$ is increasingly approaching zero as $p$ grows. Therefore, any terms involving the inner product between $a$ and $b$ can be ignored. However, another critical prerequisite needs to be additionally satisfied for the success of rectification: $[a_{t_1-p}, \cdots, a_{t_1}]$ should be closer to $[a_{t_2-p}, \cdots, a_{t_2}]$ than $[a_{t_3-p}, \cdots, a_{t_3}]$ provided that $a_{t_1} = a_{t_2}$ and $a_{t_1} \neq a_{t_3}$. This also applies to the $b$-component. For components showing stable and consistent behaviors, such as long-term and seasonal components, this prerequisite readily holds for arbitrarily large patch sizes. In contrast, for volatile components, such as the short-term component, the patch is likely to contain increasingly irrelevant information as $p$ grows, resulting in an adversely increased disparity between $x_1$ and $x_2$. Therefore, patching-based methods fall short in capturing the short-term component.

By using the decomposition mapping $D : a_{t_i} + b_{t_i} \to (a_{t_i}, b_{t_i})$, we have $\|D(x_{t_1}) - D(x_{t_2})\| = \|(0, b_{t_1} - b_{t_2})\| \leq \|(a_{t_1} - a_{t_3}, b_{t_1} - b_{t_2})\| = \|D(x_{t_1}) - D(x_{t_2})\|$. Therefore, we obtain $\|D(x_{t_1}) - D(x_{t_2})\| \leq \|D(x_{t_1}) - D(x_{t_3})\|$. We now extend our analysis to the case where the time series $x$ is sampled from different data distributions, e.g., Gaussian mixture models. Specifically, we consider a triplet $(x_{t_1}, x_{t_2}, x_{t_3})$, where $x_{t_i}$ is sampled from the distribution $\lambda_1 \mathcal{N}(\mu_{a,i}, \sigma_a I) + \lambda_2 \mathcal{N}(\mu_{b,i}, \sigma_b I)$. Here, the Gaussian distributions $\mathcal{N}(\mu_{a,i}, \sigma_a I)$ and $\mathcal{N}(\mu_{b,i}, \sigma_b I)$ represent two different components for $x_{t_i}$, and $\lambda_i$ represents the weights for the Gaussian mixture model. The goal of the decomposition approach is to decouple each component. Therefore, the decomposition projection will take the form of $D : \lambda_1 \mathcal{N}_a + \lambda_2 \mathcal{N}_b \to (\lambda_1 \mathcal{N}_a, \lambda_2 \mathcal{N}_b)^\top$. This decomposition projection can adjust the structure of the representation space and better separate data with different distributions. To show this, we suppose that $\mu_{a,1} = \mu_{a,2}$ holds. Then for any $x_3$ sampled from $\lambda_1 \mathcal{N}(\mu_{a,3}, \sigma_a I) + \lambda_2 \mathcal{N}(\mu_{b,3}, \sigma_b I)$ where $\mu_{b,3} = \mu_{b,2}$ holds, we expect that the distance between $x_1$ and $x_2$ is more likely to be closer than the distance between $x_1$ and $x_3$, since $x_1$ and $x_2$ have the same distribution component. However, the original representation cannot guarantee that the inequality $\mathbb{E}\|x_1 - x_2\|^2 \leq \mathbb{E}\|x_3 - x_1\|^2$

holds. By using the decomposition mapping, $x_{t_i}$ is mapped to the vector $(\lambda_1 a_{t_i}, \lambda_2 b_{t_i})^\top$, where $a_{t_i}$ and $b_{t_i}$ follow the distributions $\mathcal{N}(\mu_{a,i}, \sigma_a I)$ and $\mathcal{N}(\mu_{b,i}, \sigma_b I)$, respectively. Then we have $\mathbb{E}\|D(x_{t_1}) - D(x_{t_2})\|^2 = \lambda_1^2 \mathbb{E}\|a_{t_1} - a_{t_2}\|^2 + \lambda_2^2 \mathbb{E}\|b_{t_1} - b_{t_2}\|^2$. Since $\mu_{a,1} = \mu_{a,2}$, we have $\mathbb{E}\|a_{t_1} - a_{t_2}\|^2 \leq \mathbb{E}\|a_{t_1} - a_{t_3}\|^2$. It follows that $\mathbb{E}\|D(x_{t_1}) - D(x_{t_2})\|^2 \leq \lambda_1^2 \mathbb{E}\|a_{t_1} - a_{t_3}\|^2 + \lambda_2^2 \mathbb{E}\|b_{t_1} - b_{t_3}\|^2 = \mathbb{E}\|D(x_{t_1}) - D(x_{t_3})\|^2$. Therefore, the decomposition mapping ensures that the inequality $\mathbb{E}\|D(x_1) - D(x_2)\|^2 \leq \mathbb{E}\|D(x_1) - D(x_3)\|^2$ holds. This helps rectify the incorrect relations between data.

# B  Data Details

## B.1  Dataset Description

We evaluate the performance of our proposed SSCNN on 7 popular datasets with diverse regularities, including Weather, Traffic, Electricity and 4 ETT datasets (ETTh1, ETTh2, ETTm1, ETTm2). Among these 7 datasets, Traffic and Electricity consistently show more regular patterns over time, while the rest datasets contain more volatile data. Weather dataset collects 21 meteorological indicators in Germany, such as humidity and air temperature. Traffic dataset records the road occupancy rates from different sensors on San Francisco freeways. Electricity is a dataset that describes 321 customers' hourly electricity consumption. ETT (Electricity Transformer Temperature) datasets are collected from two different electric transformers labeled with 1 and 2, and each of them contains 2 different resolutions (15 minutes and 1 hour) denoted with m and h. Thus, in total we have 4 ETT datasets: ETTm1, ETTm2, ETTh1, and ETTh2. These datasets have been extensively utilized for benchmarking and publicly available on Time-Series-Library[3].

## B.2  Data Preprocessing

For the datasets with a 1-hour sampling rate—Electricity, Traffic, ETTh1, and ETTh2—the input length $T_{\text{in}}$ is uniformly set at 168 for all models. Conversely, for datasets with a 10-minute or 15-minute sampling rate, $T_{\text{in}}$ is adjusted to 432 or 384, respectively, to ensure coverage of multiple periods. The output length varies among {3, 24, 96, 192, 336}, accommodating both short-term and long-term forecasts. *To ensure fairness in the comparison, all competing methods adopt this window size configuration.* Additionally, data are pre-processed by normalization and post-processed by de-normalization across all baselines.

# C  Full Results

The full multivariate forecasting results are provided in the following section due to the space limitation of the main text. We extensively evaluate competitive counterparts on challenging forecasting tasks. Table 2 contains the detailed results of all prediction lengths of the seven well-acknowledged forecasting benchmarks. The proposed model achieves comprehensive state-of-the-art in real-world forecasting applications. We also report the standard deviation of SSCNN performance under five runs with different random seeds in Table 3, which exhibits that the performance of SSCNN is stable.

# D  Implications of Decomposition on Spatial-Temporal Correlations

We showcase that decomposition facilitates to wipe out redundant correlations, leading to sparse temporal and spatial correlation maps. We first define two concepts, namely the conditional correlation and conditional auto-correlation. Then, we utilize these two concepts to analyze the temporal correlations and spatial correlations, respectively.

## D.1  Conditional Correlation and Conditional Auto-correlation

Conditional correlation, or also known as partial correlation, refers to the degree of association between two series, with the effect of a set of controlling series removed, while conditional auto-correlation refers to the correlation of a series with a delayed copy of itself, given a specific set of controlling series.

---

[3]https://github.com/thuml/Time-Series-Library

Table 2: Long-term forecasting results on 7 real-world datasets in MSE and MAE. The best result is highlighted in **bold**, and the second best is highlighted with underline.

| Models | | SSCNN (Ours) | | SCNN (2024) | | iTransformer (2024) | | TimeMixer (2024) | | PatchTST (2023) | | TimesNet (2023) | | Crossformer (2023) | | DLinear (2023) | | Autoformer (2021) | |
|---|---|---|---|---|---|---|---|---|---|---|---|---|---|---|---|---|---|---|---|
| Metric | | MSE | MAE | MSE | MAE | MSE | MAE | MSE | MAE | MSE | MAE | MSE | MAE | MSE | MAE | MSE | MAE | MSE | MAE |
| ELC | 3 | **0.057** | **0.149** | 0.059 | 0.152 | 0.059 | 0.152 | 0.060 | 0.153 | 0.063 | 0.160 | 0.119 | 0.232 | 0.058 | 0.151 | 0.077 | 0.175 | 0.147 | 0.273 |
| | 24 | **0.093** | **0.187** | 0.096 | 0.192 | 0.094 | 0.189 | 0.097 | 0.191 | 0.100 | 0.197 | 0.135 | 0.245 | 0.098 | 0.195 | 0.122 | 0.221 | 0.168 | 0.286 |
| | 96 | **0.128** | **0.221** | 0.148 | 0.241 | 0.133 | 0.229 | 0.134 | 0.226 | 0.136 | 0.230 | 0.169 | 0.272 | 0.136 | 0.238 | 0.154 | 0.248 | 0.186 | 0.301 |
| | 192 | **0.151** | **0.243** | 0.160 | 0.252 | 0.154 | 0.247 | 0.152 | 0.244 | 0.152 | 0.244 | 0.191 | 0.288 | 0.158 | 0.255 | 0.168 | 0.260 | 0.218 | 0.328 |
| | 336 | **0.165** | **0.261** | 0.181 | 0.274 | 0.169 | 0.263 | 0.171 | 0.262 | 0.172 | 0.264 | 0.203 | 0.300 | 0.173 | 0.268 | 0.186 | 0.277 | 0.238 | 0.351 |
| Traffic | 3 | **0.241** | **0.189** | 0.246 | 0.194 | 0.250 | 0.197 | 0.265 | 0.198 | 0.252 | 0.195 | 0.510 | 0.283 | 0.289 | 0.210 | 0.331 | 0.255 | 0.524 | 0.344 |
| | 24 | **0.310** | **0.223** | 0.316 | 0.234 | 0.316 | 0.234 | 0.343 | 0.246 | 0.323 | 0.229 | 0.531 | 0.293 | 0.335 | 0.231 | 0.402 | 0.281 | 0.548 | 0.335 |
| | 96 | **0.358** | **0.244** | 0.386 | 0.271 | 0.364 | 0.258 | 0.407 | 0.278 | 0.371 | 0.251 | 0.602 | 0.319 | 0.392 | 0.272 | 0.452 | 0.302 | 0.603 | 0.368 |
| | 192 | **0.384** | **0.258** | 0.416 | 0.280 | 0.389 | 0.270 | 0.452 | 0.315 | 0.394 | 0.260 | 0.615 | 0.321 | 0.423 | 0.269 | 0.465 | 0.304 | 0.628 | 0.385 |
| | 336 | **0.398** | **0.264** | 0.437 | 0.293 | **0.398** | 0.274 | 0.472 | 0.321 | 0.409 | 0.273 | 0.640 | 0.355 | 0.440 | 0.299 | 0.480 | 0.311 | 0.616 | 0.371 |
| ETTh1 | 3 | 0.144 | **0.239** | 0.146 | 0.242 | 0.165 | 0.262 | 0.145 | 0.245 | 0.148 | 0.248 | 0.272 | 0.337 | **0.142** | 0.241 | 0.170 | 0.260 | 0.299 | 0.382 |
| | 24 | **0.297** | **0.344** | 0.304 | 0.353 | 0.320 | 0.367 | 0.299 | 0.350 | 0.299 | 0.355 | 0.352 | 0.393 | 0.318 | 0.366 | 0.319 | 0.362 | 0.442 | 0.466 |
| | 96 | **0.363** | **0.386** | 0.379 | 0.398 | 0.388 | 0.407 | 0.388 | 0.400 | 0.369 | 0.391 | 0.402 | 0.421 | 0.381 | 0.405 | 0.383 | 0.401 | 0.456 | 0.469 |
| | 192 | **0.404** | **0.413** | 0.427 | 0.423 | 0.446 | 0.444 | 0.443 | 0.444 | 0.413 | 0.425 | 0.464 | 0.459 | 0.433 | 0.431 | 0.429 | 0.428 | 0.505 | 0.491 |
| | 336 | **0.435** | 0.428 | 0.439 | **0.424** | 0.476 | 0.457 | 0.476 | 0.444 | 0.448 | 0.441 | 0.506 | 0.482 | 0.471 | 0.451 | 0.464 | 0.444 | 0.560 | 0.522 |
| ETTh2 | 3 | **0.079** | 0.177 | **0.079** | 0.177 | 0.088 | 0.193 | 0.082 | 0.179 | 0.081 | 0.178 | 0.119 | 0.232 | **0.079** | **0.176** | 0.083 | 0.182 | 0.203 | 0.310 |
| | 24 | 0.165 | 0.258 | **0.163** | **0.253** | 0.187 | 0.278 | 0.174 | 0.266 | 0.176 | 0.264 | 0.210 | 0.301 | 0.180 | 0.271 | 0.167 | 0.261 | 0.318 | 0.393 |
| | 96 | **0.285** | **0.339** | 0.289 | 0.340 | 0.304 | 0.355 | 0.302 | 0.355 | 0.300 | 0.348 | 0.340 | 0.379 | 0.328 | 0.376 | 0.290 | 0.349 | 0.378 | 0.417 |
| | 192 | **0.356** | **0.385** | 0.356 | 0.388 | 0.378 | 0.401 | 0.361 | 0.392 | 0.359 | 0.390 | 0.402 | 0.417 | 0.396 | 0.416 | 0.389 | 0.416 | 0.437 | 0.452 |
| | 336 | **0.392** | 0.419 | 0.395 | 0.417 | 0.417 | 0.428 | 0.405 | 0.429 | 0.404 | 0.434 | 0.451 | 0.454 | 0.438 | 0.430 | 0.484 | 0.482 | 0.493 | 0.486 |
| ETTm1 | 3 | 0.057 | 0.150 | 0.058 | 0.151 | 0.062 | 0.161 | 0.069 | 0.175 | **0.056** | **0.149** | 0.067 | 0.168 | 0.057 | 0.151 | 0.062 | 0.156 | 0.227 | 0.315 |
| | 24 | **0.185** | **0.265** | 0.193 | 0.270 | 0.211 | 0.290 | 0.204 | 0.284 | 0.196 | 0.277 | 0.201 | 0.282 | 0.209 | 0.282 | 0.214 | 0.288 | 0.466 | 0.446 |
| | 96 | 0.290 | **0.339** | 0.287 | 0.339 | 0.305 | 0.354 | 0.314 | 0.355 | 0.293 | 0.342 | 0.324 | 0.370 | 0.319 | 0.355 | 0.308 | 0.358 | 0.471 | 0.445 |
| | 192 | **0.323** | **0.362** | 0.327 | 0.366 | 0.346 | 0.377 | 0.331 | 0.373 | 0.335 | 0.369 | 0.393 | 0.413 | 0.374 | 0.390 | 0.341 | 0.373 | 0.566 | 0.498 |
| | 336 | **0.357** | **0.384** | 0.358 | 0.385 | 0.381 | 0.397 | 0.373 | 0.396 | 0.364 | 0.389 | 0.371 | 0.399 | 0.405 | 0.414 | 0.372 | 0.388 | 0.594 | 0.523 |
| ETTm2 | 3 | **0.041** | **0.117** | 0.042 | 0.119 | 0.044 | 0.127 | 0.043 | 0.122 | 0.042 | 0.120 | 0.051 | 0.143 | 0.042 | 0.120 | 0.044 | 0.125 | 0.120 | 0.234 |
| | 24 | **0.094** | **0.190** | 0.095 | 0.192 | 0.102 | 0.200 | 0.097 | 0.195 | **0.093** | 0.191 | 0.108 | 0.210 | 0.098 | 0.197 | 0.098 | 0.198 | 0.151 | 0.262 |
| | 96 | 0.165 | 0.254 | **0.163** | **0.250** | 0.188 | 0.274 | 0.170 | 0.258 | 0.171 | 0.255 | 0.192 | 0.278 | 0.177 | 0.264 | 0.177 | 0.273 | 0.231 | 0.317 |
| | 192 | 0.221 | 0.293 | **0.221** | **0.292** | 0.244 | 0.312 | 0.229 | 0.300 | 0.233 | 0.296 | 0.241 | 0.315 | 0.231 | 0.303 | 0.228 | 0.306 | 0.288 | 0.343 |
| | 336 | **0.270** | 0.326 | 0.271 | **0.326** | 0.299 | 0.345 | 0.282 | 0.333 | 0.296 | 0.339 | 0.309 | 0.345 | 0.292 | 0.339 | 0.291 | 0.353 | 0.339 | 0.377 |
| Weather | 3 | **0.044** | **0.060** | 0.046 | 0.066 | 0.046 | 0.062 | 0.045 | 0.062 | 0.045 | 0.062 | 0.055 | 0.091 | 0.045 | 0.064 | 0.048 | 0.074 | 0.054 | 0.087 |
| | 24 | **0.087** | **0.116** | 0.089 | 0.120 | 0.097 | 0.130 | 0.092 | 0.133 | 0.093 | 0.121 | 0.100 | 0.142 | 0.093 | 0.134 | 0.102 | 0.147 | 0.119 | 0.167 |
| | 96 | 0.142 | 0.196 | **0.142** | **0.192** | 0.168 | 0.216 | 0.145 | 0.198 | 0.163 | 0.207 | 0.173 | 0.221 | 0.155 | 0.212 | 0.171 | 0.224 | 0.201 | 0.242 |
| | 192 | **0.186** | **0.231** | 0.186 | 0.234 | 0.213 | 0.252 | 0.191 | 0.240 | 0.203 | 0.243 | 0.216 | 0.266 | 0.194 | 0.247 | 0.219 | 0.282 | 0.253 | 0.311 |
| | 336 | **0.238** | **0.274** | 0.239 | 0.276 | 0.269 | 0.294 | 0.243 | 0.280 | 0.262 | 0.287 | 0.285 | 0.314 | 0.264 | 0.317 | 0.266 | 0.321 | 0.302 | 0.350 |

Table 3: Robustness of SSCNN performance. The results are obtained from five random seeds.

| Dataset | ECL | | Traffic | | ETTh1 | |
|---|---|---|---|---|---|---|
| Metrics | MSE | MAE | MSE | MAE | MSE | MAE |
| 3 | 0.057±0.000 | 0.149±0.000 | 0.241±0.003 | 0.189±0.001 | 0.144±0.004 | 0.239±0.003 |
| 24 | 0.093±0.001 | 0.187±0.002 | 0.310±0.004 | 0.223±0.003 | 0.297±0.003 | 0.344±0.003 |
| 96 | 0.128±0.002 | 0.221±0.003 | 0.358±0.002 | 0.244±0.002 | 0.363±0.002 | 0.386±0.004 |
| 192 | 0.151±0.000 | 0.243±0.001 | 0.384±0.004 | 0.258±0.002 | 0.404±0.003 | 0.413±0.005 |
| 336 | 0.165±0.003 | 0.261±0.003 | 0.398±0.002 | 0.264±0.002 | 0.435±0.003 | 0.428±0.004 |
| Datasets | ETTh2 | | ETTm1 | | ETTm2 | |
| Metrics | MSE | MAE | MSE | MAE | MSE | MAE |
| 3 | 0.079±0.000 | 0.177±0.001 | 0.057±0.001 | 0.150±0.002 | 0.041±0.002 | 0.117±0.002 |
| 24 | 0.165±0.001 | 0.258±0.002 | 0.185±0.002 | 0.265±0.003 | 0.094±0.001 | 0.190±0.001 |
| 96 | 0.285±0.001 | 0.339±0.001 | 0.290±0.003 | 0.339±0.001 | 0.165±0.001 | 0.254±0.002 |
| 192 | 0.356±0.004 | 0.385±0.004 | 0.323±0.003 | 0.362±0.004 | 0.221±0.001 | 0.293±0.002 |
| 336 | 0.392±0.004 | 0.419±0.002 | 0.357±0.003 | 0.384±0.001 | 0.270±0.002 | 0.326±0.003 |

**Definition 1** (Conditional Correlation). Let $X$ and $Y$ be two real-valued, one-dimensional series, and $\mathbf{Z}$ an $n$-dimensional control series. Denote $X_i$, $Y_i$, and $\mathbf{Z}i$ as the $i^{\text{th}}$ observations of each series, respectively. The conditional correlation is defined based on the residuals in $X$ and $Y$ that are unexplained by $\mathbf{Z}$. The residuals are calculated as follows:

$$R_{X,i} = X_i - \mathbf{w}_X \cdot \mathbf{z}_i,$$
$$R_{Y,i} = Y_i - \mathbf{w}_Y \cdot \mathbf{z}_i,$$

where $\mathbf{w}_X$ and $\mathbf{w}_Y$ are obtained by minimizing the respective squared differences:

$$\mathbf{w}_X = \arg\min_{\mathbf{w}} \left\{ \sum_{i=1}^{N} (X_i - \mathbf{w} \cdot \mathbf{Z}_i)^2 \right\},$$

$$\mathbf{w}_Y = \arg\min_{\mathbf{w}} \left\{ \sum_{i=1}^{N} (Y_i - \mathbf{w} \cdot \mathbf{Z}_i)^2 \right\}.$$

The conditional correlation $\hat{\rho}_{XY|\mathbf{Z}}$ is then computed using these residuals:

$$\hat{\rho}_{XY|\mathbf{Z}} = \frac{\sum_{i=1}^{N} R_{X,i} R_{Y,i}}{\sqrt{\sum_{i=1}^{N} R_{X,i}^2} \sqrt{\sum_{i=1}^{N} R_{Y,i}^2}}.$$

**Definition 2** (Conditional Auto-correlation). Extending the concept of conditional correlation to auto-correlation, we analyze a series' correlation with its own past values in the context of other variables. For a time series $Y$ and control variables $\mathbf{Z}$, the residual is computed as:

$$R_i = Y_i - \mathbf{w} \cdot \mathbf{Z}_i,$$

where $\mathbf{w}$ is determined by minimizing the squared differences:

$$\mathbf{w} = \arg\min_{\mathbf{w}} \left\{ \sum_{i=1}^{N} (Y_i - \mathbf{w} \cdot \mathbf{Z}_i)^2 \right\}.$$

The conditional auto-correlation is then defined as:

$$\hat{\rho}_{Y|\mathbf{Z}}(\tau) = \frac{\sum_{i=1+\tau}^{N} R_i R_{i-\tau}}{\sqrt{\sum_{i=1}^{N} R_i^2} \sqrt{\sum_{i=1+\tau}^{N} R_{i-\tau}^2}}.$$

Here, $\tau$ represents the time lag. The measure $\hat{\rho}_{YY|\mathbf{Z}}(\tau)$ quantifies the correlation between the time series and its $\tau$-lagged series, conditional on $\mathbf{Z}$.

Both conditional correlation and conditional auto-correlation offer flexible and diverse instantiation possibilities, relying on the definition of the conditional set. When this set is null, these measures simplify to their standard forms of correlation and auto-correlation, respectively. This adaptability allows for a broad range of analytical applications and interpretations. However, when utilizing conditional auto-correlation for time series data analysis, we encounter a significant challenge: the limited information of the controlling variable $\mathbf{Z}$. In practice, identifying and evaluating the factors influencing our time series often requires external data sources, which are not always available. Instead, we use the method proposed by SSCNN to approximate the components that can be explained by factors with specific regularities. Figure 6 showcases the estimated components alongside the residuals obtained by incrementally controlling the long-term (e.g., resulting from resident population), seasonal (e.g., resulting from time of day), and short-term (e.g., resulting from occurrence of an event) components.

### D.2 Temporal Regularity

Temporal regularity can be captured by the evolution of conditional auto-correlation, as depicted in Fig. 7 (a)(b)(c)(d), indicating how the structured components impact the temporal regularity of time series. We can observe that the original time series degenerates into a sequence of nearly uncorrelated residuals upon progressively controlling for long-term, seasonal, and short-term components. This observation implies that these diverse components collectively elucidate the dynamics of time series, rendering any remaining correlations superfluous. Therefore, isolating these components is imperative to pinpoint and eliminate potential redundancy. Moreover, employing an operator with an extensive receptive field, such as the MLP used in ?Nie et al. (2023), becomes unnecessary for processing decoupled data due to the minimal relevance of current observations to distant lags.

In addition, conditional auto-correlations display variations with increasing lag intervals, likely attributed to cumulative noise. This fluctuation highlights the necessity for the selection mechanism, to discern subtle differences, moving beyond basic approaches like moving averages that uniformly treat all lags.

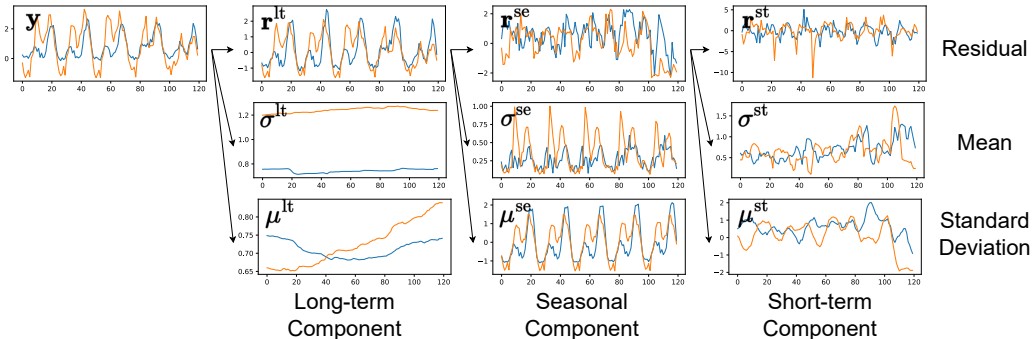

Figure 6: Disentangling structured components from the original time series data.

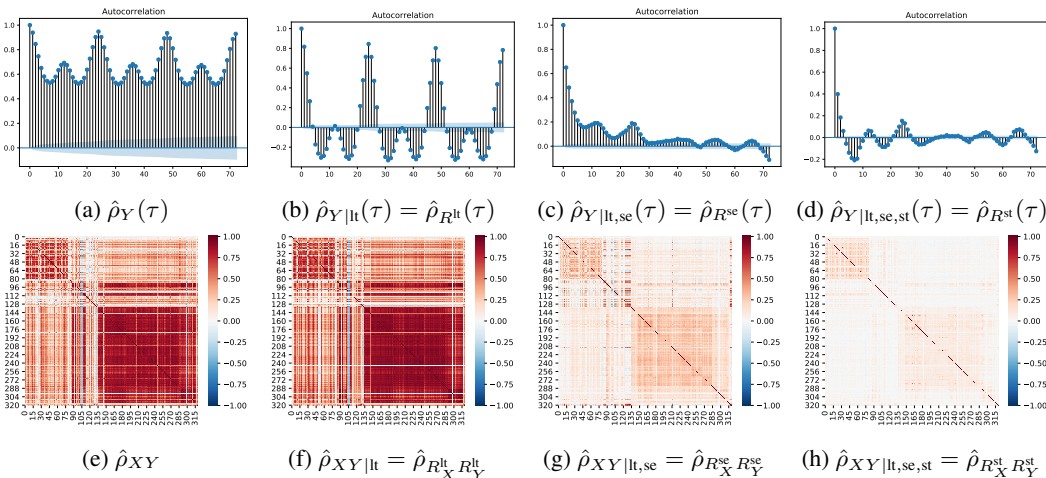

(a) $\hat{\rho}_Y(\tau)$  (b) $\hat{\rho}_{Y|\mathrm{lt}}(\tau) = \hat{\rho}_{R^{\mathrm{lt}}}(\tau)$  (c) $\hat{\rho}_{Y|\mathrm{lt},\mathrm{se}}(\tau) = \hat{\rho}_{R^{\mathrm{se}}}(\tau)$  (d) $\hat{\rho}_{Y|\mathrm{lt},\mathrm{se},\mathrm{st}}(\tau) = \hat{\rho}_{R^{\mathrm{st}}}(\tau)$

(e) $\hat{\rho}_{XY}$  (f) $\hat{\rho}_{XY|\mathrm{lt}} = \hat{\rho}_{R_X^{\mathrm{lt}} R_Y^{\mathrm{lt}}}$  (g) $\hat{\rho}_{XY|\mathrm{lt},\mathrm{se}} = \hat{\rho}_{R_X^{\mathrm{se}} R_Y^{\mathrm{se}}}$  (h) $\hat{\rho}_{XY|\mathrm{lt},\mathrm{se},\mathrm{st}} = \hat{\rho}_{R_X^{\mathrm{st}} R_Y^{\mathrm{st}}}$

Figure 7: (a)(b)(c)(d): The evolution of conditional correlation between each pair of series when progressively controlling for three distinct types of factors. (e)(f)(g)(h):The evolution of conditional auto-correlation when progressively controlling for three distinct types of factors.

### D.3 Spatial Regularity

Spatial regularity is recognized as another vital information source for enhancing prediction accuracy and reliability in many datasets. However, an long-standing question in spatial regularity research persists: Why the channel-mixing strategy struggles to capture beneficial spatial regularities Han et al. (2023b); Nie et al. (2023)?

Our principal discovery is that spatial and temporal correlations frequently intersect, leading to redundant information. This overlap necessitates the identification of distinct spatial correlations that augment temporal correlations. We concentrate on conditional correlations, acquired by methodically controlling for long-term, seasonal, and short-term components. As Fig. 7 (e)(f)(g)(h) illustrates, spatial correlations gradually wane as more temporal elements are regulated. Eventually, when all three factors are considered, the majority of series pairs exhibit no significant relation, underscoring the criticality of discerning genuinely impactful correlations.

This phenomenon is intuitively understood. Temporal influences, such as seasonal trends, commonly affect multiple time series concurrently, fostering parallel evolutions and visible spatial correlations. Once these influences are adjusted for, the resulting spatial correlations wane, uncovering less correlated series. It is the correlations among these residual series that genuinely refine forecasting by supplying unique spatial insights. Thus, the key insight for spatial modeling is that conditional correlation is instrumental in highlighting spatial variables that offer information complementary to temporal data.

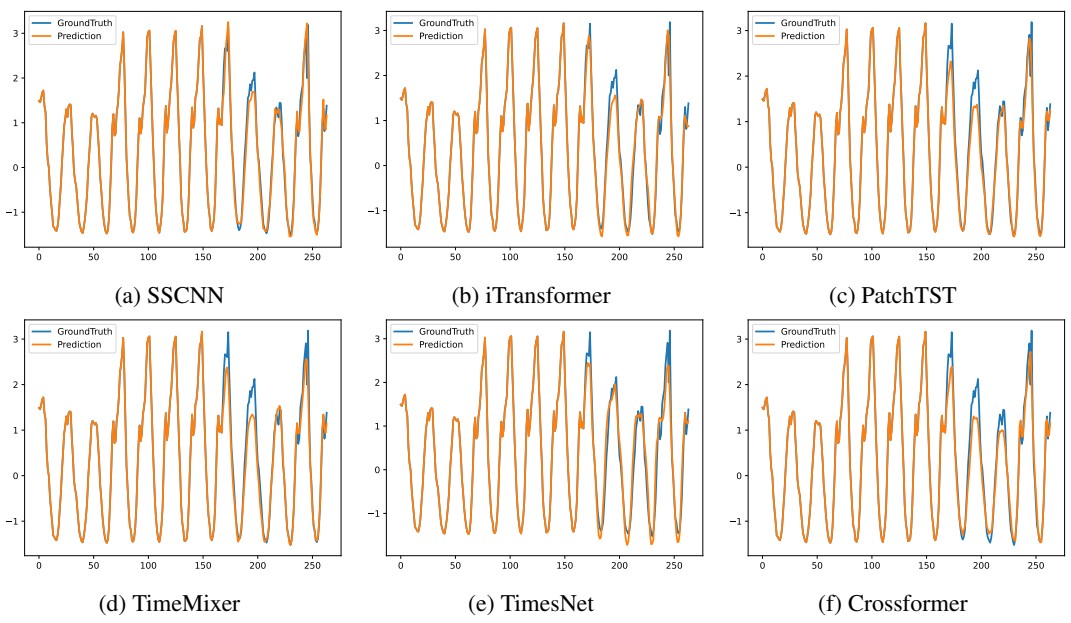

(a) SSCNN   (b) iTransformer   (c) PatchTST

(d) TimeMixer   (e) TimesNet   (f) Crossformer

Figure 8: Visualization of input-168-predict-96 results on the Traffic dataset.

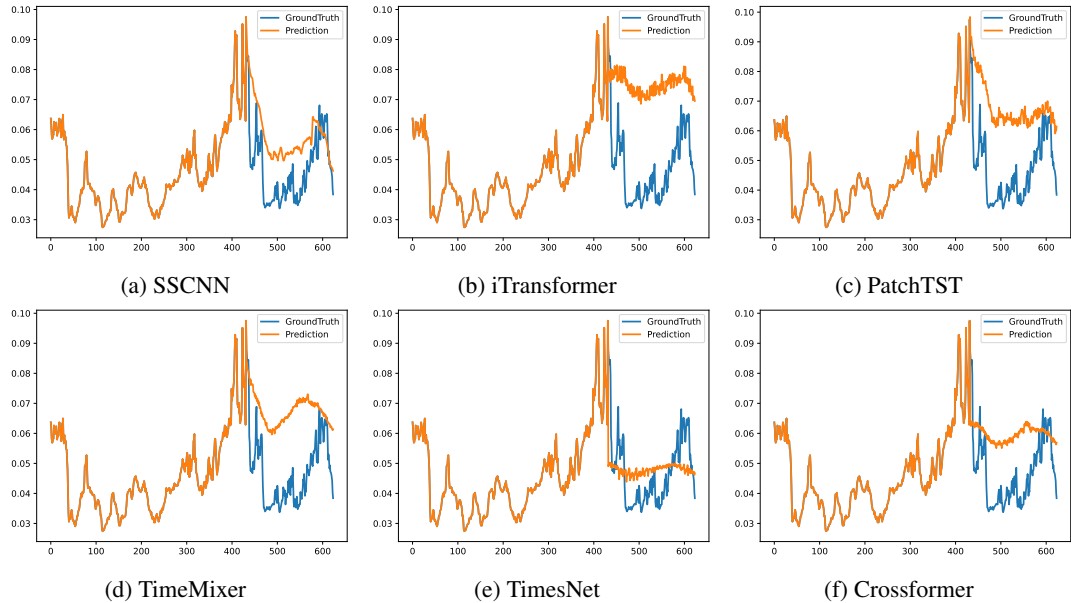

(a) SSCNN   (b) iTransformer   (c) PatchTST

(d) TimeMixer   (e) TimesNet   (f) Crossformer

Figure 9: Visualization of input-432-predict-192 results on the Weather dataset.

## E   Visualization of Prediction Results

To provide a clear comparison among different models, we list supplementary prediction showcases of two representative datasets in Fig. 8 and Fig. 9, respectively, which are given by the following models: SSCNN, iTransfomrer (**?**), PatchTST (Nie et al., 2023), Crossformer (Zhang & Yan, 2023), TimesNet (Wu et al., 2023), TimeMixer (Wang et al., 2024). Among the various models, SSCNN predicts the most precise future series variations and exhibits superior performance

