# OpenReview forum: "Parsimony or Capability? Decomposition Delivers Both in Long-term Time Series Forecasting"
_NeurIPS.cc/2024/Conference — NeurIPS 2024 spotlight_

### Official Review · Reviewer_p9WT · 2024-07-04

**Soundness:** 3
**Presentation:** 3
**Contribution:** 3
**Rating:** 7
**Confidence:** 4

**Summary:**

This paper proposes  a Selective Structured Components-based Neural Network for Long-term Time Series Forecasting

**Strengths:**

1. This paper demonstrates originality by addressing a crucial limitation in existing SOTA methods, maintains high quality through thorough experimentation and clear presentation, offers significant advancements to the field of time series forecasting, and ensures clarity that aids in understanding and reproduction of the work.

2. The motivation of this paper is intuitive and compelling. Given the large model sizes of current SOTA methods like PatchTST, the idea of using a smaller model to achieve comparable or better performance is highly attractive.

3. The experiments are thorough, and the proposed method achieves state-of-the-art performance. The effectiveness of each sub-module is demonstrated through detailed ablation studies.

4. The code is open source and reproducible, with a straightforward and clear usage process.

**Weaknesses:**

1. In the experiment section, it is noted that most papers use the ETT dataset for ablation studies, likely due to its smaller size, which allows for quicker results. However, you chose the ECL and Traffic datasets instead of ETT, which is a more comprehensive and reliable approach. While this choice is commendable, there is no explanation provided for not using the ETT dataset.

2.It would be more informative to report the model size directly in Table 1. Including the model size would provide a clearer comparison with other SOTA methods and highlight the efficiency of your proposed model.

3.Baselines: Some MTSF models based on LLM have been widely applied [1]. If the authors can demonstrate that SSCNN has advantages in both performance and efficiency, this paper will be more convincing.

4.Some extremely lightweight models have also been proven to have satisfactory performance [2] . Compared to these methods, what are the main advantages of SSCNN?

[1]One Fits All:Power General Time Series Analysis by Pretrained LM

[2]FITS: Modeling Time Series with 10k Parameters

**Questions:**

See weakness

**Limitations:**

No problem

---

> ### Author Rebuttal · Authors · 2024-08-05
>
> Thank you for your thoughtful review and constructive comments.
>
> ---
> **W1:**  We chose the ECL and Traffic datasets for our experiments primarily because they present more challenging tasks due to their larger data scale. Specifically, these datasets include a mass of variables and cover an extensive data collection period. This increased data scale results in a less biased estimate of the impact of each module, providing a more comprehensive evaluation of our model's performance. To address the concern regarding the exclusion of the ETT dataset, we have conducted an additional ablation study using the ETTh1 dataset. The results from this study are provided below for your reference. This inclusion further validates the robustness and generalizability of our approach across different dataset scales and complexities. We will improve this study in the revision.
>
>
> |SSCNN|w/o lta |  w/o lt | w/o sta | w/o st | w/o sea | w/o se | w si |
> |---|----|---|---|---|---|---|---|
> |0.363| 0.364 | 0.379 |  0.364 |  0.368 |  0.364 |  0.415|  0.378 |
>
>
> ---
> **W2:** We will include the model size in Table 1 in the final version of the paper.
>
> ---
> **W3:** We have conducted a comparative analysis between SSCNN and MTSF models based on LLM. Due to the time constraint, we directly copy the results obtained by baselines from an ICML 2024 publication [1]. The results, which are reported at the end of the reply box, demonstrate SSCNN's advantages in both performance and efficiency. This analysis strengthens our claim that SSCNN is a highly competitive model in the field of time series forecasting. We will include this comparative analysis in the final version of the paper to provide a more comprehensive evaluation of SSCNN's capabilities.
>
> ---
> **W4:** We appreciate your concern, which prompts us to emphasize the key contributions of SSCNN that might otherwise be underestimated. As discussed in the last paragraph of Section 2, many lightweight models achieve a reduction in parameters, however, at the cost of sacrificing prediction accuracy. In contrast, SSCNN manages to maintain strong competitive performance with comparably reduced parameters. To substantiate this claim, we offer an additional comparison between SSCNN and some representative lightweight models, including FITS, highlighting the balance SSCNN strikes between model complexity and prediction accuracy. Due to the time constraint, we directly borrow the results obtained by baselines from an ICML 2024 publication [2]. This comparison will be included in the supplementary materials of the final version, showcasing SSCNN's advantages in terms of both efficiency and effectiveness in time series forecasting.
>
> [1] Bian, Yuxuan, et al. "Multi-patch prediction: Adapting llms for time series representation learning." arXiv preprint arXiv:2402.04852 (2024).
>
> [2] Lin, Shengsheng, et al. "SparseTSF: Modeling Long-term Time Series Forecasting with 1k Parameters." arXiv preprint arXiv:2405.00946 (2024).
>
>
> | Electricity | 96 | 192 | 336 | 720 |
> |-----------|-----------|-----------|-------------|-------------|
> | PatchTST (Transformer-based ICLR 2023) | 0.129 | 0.147 | 0.167 | 0.202 |
> | FITS (lightweight, ICLR 2024)  |       0.138| 0.152 | 0.166 | 0.205 |
> | SparseTSF (lightweight, ICML 2024)| 0.138 | 0.146 | 0.164 | 0.203 |
> | ALLM4TS (LLM-based, ICML 2024) | 0.127 | 0.145 | 0.163 | 0.206 |
> |Time-LLM (LLM-based, ICLR 2024) | 0.140 | 0.151 | 0.171 | 0.210 |
> |SSCNN (Ours) |   **0.126** | **0.145** | **0.161** | **0.191** |
>
>
> | Traffic | 96 | 192 | 336 | 720 |
> |-----------|-----------|-----------|-------------|-------------|
> | PatchTST (Transformer-based ICLR 2023) | 0.367 | 0.385 | 0.399 | 0.434
> | ALLM4TS (LLM-based, ICML 2024) | 0.372 | 0.383 | 0.396 | 0.433
> |Time-LLM (LLM-based, ICLR 2024) | 0.383 | 0.398 | 0.407 | 0.434 |
> |FITS (lightweight, ICLR 2024)    |    0.401| 0.407 | 0.420 | 0.456 |
> |SparseTSF (lightweight, ICML 2024) | 0.382 | 0.388 | 0.402 | 0.445 |
> |SSCNN (Ours) |   **0.352** | **0.380** | **0.390** | **0.423** |
>
> | ETTh1 | 96 | 192 | 336 | 720 |
> |-----------|-----------|-----------|-------------|-------------|
> | PatchTST (Transformer-based ICLR 2023) | 0.379 | 0.413 | 0.422 | 0.447 |
> | ALLM4TS (LLM-based, ICML 2024) | 0.380 | **0.396** | 0.413 | 0.461 |
> |Time-LLM (LLM-based, ICLR 2024) | 0.399 | 0.433 | 0.469 | 0.473 |
> FITS (lightweight, ICLR 2024)    |     0.375| 0.408 | 0.429 | 0.427 |
> |SparseTSF (lightweight, ICML 2024) | **0.359** | 0.397 | **0.404** | **0.417** |
> |SSCNN (Ours) |   **0.361** | **0.401** | **0.410** | **0.424** |

---

> > ### Comment · Reviewer_p9WT · 2024-08-12
> >
> > I am very grateful for the detailed response from the author. Currently, SSCNN has achieved excellent results in both performance and efficiency. I have a small question: Were the compared LLMs (Time-LLM and ALLM4TS) evaluated using conventional training and testing methods, or were they using Few-shot and Zero-shot settings?

---

> > > ### Author Response · Authors · 2024-08-12
> > >
> > > We sincerely appreciate your feedback and the constructive suggestions you've provided for improving our paper. To answer your question, the compared LLMs were evaluated using conventional training and testing methods with the full training data. We agree that exploring SSCNN's performance under few-shot and zero-shot settings would be valuable, and we plan to investigate this in future work.

---

> > > > ### Comment · Reviewer_p9WT · 2024-08-13
> > > >
> > > > Thank you to the author for addressing all of my concerns. I believe SSCNN is a solid work. I will increase my score.

---

> > > > > ### Author Response · Authors · 2024-08-13
> > > > >
> > > > > Dear Reviewer p9WT,
> > > > >
> > > > > we sincerely value your feedback and the constructive suggestions you've provided for enhancing our paper.
> > > > >
> > > > > Authors

---

### Official Review · Reviewer_aS9R · 2024-07-08

**Soundness:** 3
**Presentation:** 3
**Contribution:** 3
**Rating:** 6
**Confidence:** 4

**Summary:**

This paper identifies data decomposition as a core bottleneck in time series forecasting and proposes a novel model named SSCNN, a decomposition-based model innovatively enhanced with a selection mechanism. SSCNN is specifically designed to adeptly capture complex regularities in data while maintaining a minimal parameter scale. This paper also provides an in-depth comparison between decomposition and patching, examining both capability and parsimony.  Comprehensive experiments show the superior performance of SSCNN.

**Strengths:**

Strong Points:

1. The insight of this paper is attractive and compelling. One of the most crucial characteristics of time series is that they can be viewed as composed of components with different natures, e.g., season, trend, and residual. However, this characteristic has been rarely utilized in related works, or it has been implemented in trivial ways. This paper identifies data decomposition as a core bottleneck in time series forecasting and proves its effectiveness. By decomposing complex data into more learnable components, SSCNN achieves state-of-the-art performance with a minimal number of parameters.

2. The writing of this paper is very clear. I can easily follow the author's logic and understand their points.

3. The experimental results are extensive, including overall performance results, ablation studies of each component, hyperparameter experiments, etc., which validate the effectiveness of SSCNN.

4. The code is reproducible and well documented. I have successfully replicated the authors' results.

5. The authors also provide an in-depth comparative analysis and experimental results between patching and decomposition, which help readers understand the advantages of SSCNN’s insights.

This paper emphasizes the importance of decomposition in long-term time series forecasting, addressing the analytical gap in feature decomposition for the first time and providing insights into its rationale for capability and parsimony compared to patching.

**Weaknesses:**

I have some minor questions and suggestions. If the author addresses the following points, I will increase my score.

Weak Points:

Experimental Setting: Most works using the Time-Series-Library repository predict up to 720 steps, yet your results do not include this prediction horizon. It would be beneficial to explain why 720-step predictions were not included.

Figures: I suggest the authors add more explanatory information to Figure 1 to help readers grasp the main architecture of SSCNN from the figure and its caption alone. Moreover, some font styles (italic) in Figure 1 seem different from the character styles in the main text. I recommend unifying the styles.

Minor Issues: The operator $\lfloor \cdot \rfloor$ is used in the paper but not explained. In Figure 3(a), if I understand correctly, “HDformer” should be replaced by “SSCNN.”

Figures: The text size of the legends in the figures is too small, making them difficult to read. Adjusting the text size to be consistent with the main text would enhance the readability of the figures and improve the overall presentation quality of the paper.

**Questions:**

See Weakness.

**Limitations:**

Yes

---

> ### Author Rebuttal · Authors · 2024-08-05
>
> Thank you for your thoughtful review and constructive comments.
>
> ---
> **W1:** We acknowledge the importance of including 720-step predictions. In our experiments, we observed that the ranking of models for 720 output steps differed only slightly from the ranking for 336 output steps. This suggests that the models maintain consistent performance over long-term horizons. To support this observation, we conducted additional experiments with SSCNN and selected baselines on the ECL, Traffic, and ETTh1 datasets for output steps within 96, 192, 336 and 720.
>
> | Electricity | 96 | 192 | 336 | 720 |
> |-----------|-----------|-----------|-------------|-------------|
> | PatchTST (Transformer-based ICLR 2023) | 0.129 | 0.147 | 0.167 | 0.202 |
> | FITS (lightweight, ICLR 2024)  |       0.138| 0.152 | 0.166 | 0.205 |
> | SparseTSF (lightweight, ICML 2024)| 0.138 | 0.146 | 0.164 | 0.203 |
> | ALLM4TS (LLM-based, ICML 2024) | 0.127 | 0.145 | 0.163 | 0.206 |
> |Time-LLM (LLM-based, ICLR 2024) | 0.140 | 0.151 | 0.171 | 0.210 |
> |SSCNN (Ours) |   **0.126** | **0.145** | **0.161** | **0.191** |
>
>
> | Traffic | 96 | 192 | 336 | 720 |
> |-----------|-----------|-----------|-------------|-------------|
> | PatchTST (Transformer-based ICLR 2023) | 0.367 | 0.385 | 0.399 | 0.434
> | ALLM4TS (LLM-based, ICML 2024) | 0.372 | 0.383 | 0.396 | 0.433
> |Time-LLM (LLM-based, ICLR 2024) | 0.383 | 0.398 | 0.407 | 0.434 |
> |FITS (lightweight, ICLR 2024)    |    0.401| 0.407 | 0.420 | 0.456 |
> |SparseTSF (lightweight, ICML 2024) | 0.382 | 0.388 | 0.402 | 0.445 |
> |SSCNN (Ours) |   **0.352** | **0.380** | **0.390** | **0.423** |
>
> | ETTh1 | 96 | 192 | 336 | 720 |
> |-----------|-----------|-----------|-------------|-------------|
> | PatchTST (Transformer-based ICLR 2023) | 0.379 | 0.413 | 0.422 | 0.447 |
> | ALLM4TS (LLM-based, ICML 2024) | 0.380 | **0.396** | 0.413 | 0.461 |
> |Time-LLM (LLM-based, ICLR 2024) | 0.399 | 0.433 | 0.469 | 0.473 |
> FITS (lightweight, ICLR 2024)    |     0.375| 0.408 | 0.429 | 0.427 |
> |SparseTSF (lightweight, ICML 2024) | **0.359** | 0.397 | **0.404** | **0.417** |
> |SSCNN (Ours) |   **0.361** | **0.401** | **0.410** | **0.424** |
>
> ---
> **W2:**  To provide a clearer understanding of the SSCNN architecture, we will enhance Figure 1 with more explanatory information. Here's an overview: (a) Embedding: Raw data points are independently mapped to a high-dimensional representation space using a shared 1x1 convolution operator across all variables and horizons. (b) In-sample Inference: The historical time series representations are processed by three temporal attention-based normalization (T-AttnNorm) layers with different instantiations of the attention map ($\mathcal{I}$), followed by a spatial attention-based normalization (S-AttnNorm) layer. This process yields estimations of four structured components ($\mu$) along with the residual ($R$). (c) Out-of-sample extrapolation: The decomposed components corresponding to each variable are individually extrapolated to future horizons using the matrix multiplication (MatMul) operator, with distinct instantiations of the attention map ($\mathcal{E}$). (d) Component fusion: Finally, all four components, combined with the residuals, are input into a polynomial regression layer to capture their complex interrelations.
>
> Additionally, we will ensure that the font styles in Figure 1 are consistent with the main text in the final version.
>
> ---
> **W3:** We appreciate your attention to these issues. To address them:
>
> (a) We will include a clear definition and explanation of the operator $\lfloor . \rfloor$, which denotes the floor function, in the paper.
>
> (b) Regarding Figure 3(a), we will correct the labeling error by replacing “HDformer” with “SSCNN”.
>
> (c) We acknowledge the issue with the text size of the legends in the figures. We will adjust the text size to ensure consistency with the main text, thereby enhancing the readability and overall presentation quality of the paper.
>
> Thank you for your helpful feedback.

---

> > ### Comment · Reviewer_aS9R · 2024-08-13
> >
> > Thanks for your detailed response. My concerns have been well addressed.

---

> > > ### Author Response · Authors · 2024-08-13
> > >
> > > Dear Reviewer aS9R,
> > >
> > > we sincerely value your feedback and the constructive suggestions you've provided for enhancing our paper. If you have any further questions or concerns, please feel free to let us know.
> > >
> > > Authors

---

### Official Review · Reviewer_aFLy · 2024-07-08

**Soundness:** 3
**Presentation:** 3
**Contribution:** 3
**Rating:** 7
**Confidence:** 4

**Summary:**

This paper addresses long-term time series forecasting and critiques the reliance on complex models with extensive parameters. It proposes a decomposition method specifically designed for time series dynamics, achieving better forecasting performance across various datasets. Remarkably, the new model uses over 99% fewer parameters than other methods, highlighting the efficiency of domain-specific approaches. This research calls for a move away from complexity in LTSF, showcasing the effectiveness of focused decomposition techniques rather than relying on large-scale models.

**Strengths:**

1.	The paper is praiseworthy for its intuitive approach. It tackles a significant problem by proposing a method that matches or surpasses current state-of-the-art models like PatchTST while using a smaller model footprint. The experimental results strongly validate this approach.

2.	The model consistently performs well under various experimental conditions, including different input window sizes and hyperparameter settings. Statistical tests demonstrate the reliability of the results across multiple initializations, strengthening the study's credibility.

3.	The authors provide a thorough comparison between decomposition and patching in terms of effectiveness and simplicity, demonstrating the superior benefits of decomposition over patching.

**Weaknesses:**

1.	The clarity of the methodology could be improved with further elaboration.
2.  The evaluation could be strengthened by including comparisons with LLM-based models, such as:

 [1] Jin, Ming, et al. "Time-LLM: Time Series Forecasting by Reprogramming Large Language Models." The Twelfth International Conference on Learning Representations.

[2] Bian, Yuxuan, et al. "Multi-patch prediction: Adapting llms for time series representation learning." arXiv preprint arXiv:2402.04852 (2024).

**Questions:**

1.	The methodology would benefit from additional explanation regarding the structure and rationale of the Polynomial Regression layer depicted in Figure 1. If this layer represents a standard approach, including references would enhance clarity.

2.	Clarifying the decision to exclude an attention mechanism from the long-term component, despite its presence in other components like seasonal, short-term, and spatial, would strengthen the methodological coherence and aid reader comprehension.

3.	Figure 1 requires clarification on several elements. Specifically, the purpose of the 4x4 blocks and addressing inconsistent text formatting (e.g., $\mathcal{E}$) compared to the main text would improve comprehensibility.

**Limitations:**

Whether this model can be applied to other types of time series data, e.g. trajectory.

---

> ### Author Rebuttal · Authors · 2024-08-05
>
> Thank you for your thoughtful review and constructive comments.
>
> ---
> **W1:** We have addressed the concerns raised in your questions. Please kindly refer to the response to Q1 and Q2 below.
>
> ---
> **W2:**  We have conducted a comparative analysis between SSCNN and MTSF models based on LLM [1] [2]. Due to the time constraint, we directly copy the results obtained by baselines from an ICML 2024 publication [1]. The results, which are reported at the end of the reply box, demonstrate SSCNN's advantages in both performance and efficiency. This analysis strengthens our claim that SSCNN is a highly competitive model in the field of time series forecasting. We will include this comparative analysis in the final version of the paper to provide a more comprehensive evaluation of SSCNN's capabilities.
>
> [1] Bian, Yuxuan, et al. "Multi-patch prediction: Adapting llms for time series representation learning." arXiv preprint arXiv:2402.04852 (2024).
>
> [2] Jin, Ming, et al. "Time-LLM: Time Series Forecasting by Reprogramming Large Language Models." The Twelfth International Conference on Learning Representations.
>
> | Electricity | 96 | 192 | 336 | 720 |
> |-----------|-----------|-----------|-------------|-------------|
> | PatchTST (Transformer-based ICLR 2023) | 0.129 | 0.147 | 0.167 | 0.202 |
> | ALLM4TS (LLM-based, ICML 2024) | 0.127 | 0.145 | 0.163 | 0.206
> |Time-LLM (LLM-based, ICLR 2024) | 0.140 | 0.151 | 0.171 | 0.210 |
> |SSCNN (Ours) |   **0.126** | **0.145** | **0.161** | **0.191** |
>
>
> | Traffic | 96 | 192 | 336 | 720 |
> |-----------|-----------|-----------|-------------|-------------|
> | PatchTST (Transformer-based ICLR 2023) | 0.367 | 0.385 | 0.399 | 0.434
> | ALLM4TS (LLM-based, ICML 2024) | 0.372 | 0.383 | 0.396 | 0.433
> |Time-LLM (LLM-based, ICLR 2024) | 0.383 | 0.398 | 0.407 | 0.434 |
> |SSCNN (Ours) |   **0.352** | **0.380** | **0.390** | **0.423** |
>
> | ETTh1 | 96 | 192 | 336 | 720 |
> |-----------|-----------|-----------|-------------|-------------|
> | PatchTST (Transformer-based ICLR 2023) | 0.379 | 0.413 | 0.422 | 0.447 |
> | ALLM4TS (LLM-based, ICML 2024) | 0.380 | **0.396** | 0.413 | 0.461 |
> |Time-LLM (LLM-based, ICLR 2024) | 0.399 | 0.433 | 0.469 | 0.473 |
> |SSCNN (Ours) |   **0.361** | 0.401 | **0.410** | **0.424** |
>
> ---
> **Q1:** The Polynomial Regression layer depicted in Figure 1 is inspired by the work of [3], where we extend the original module to include both additive and multiplicative relations. This extension allows us to model more complex interactions between the decomposed components. We will provide a more detailed explanation of the Polynomial Regression layer in the methodology section, including its structure and rationale. Additionally, we will cite relevant references to enhance the clarity and contextual understanding of this approach.
>
> [3] Deng, Jinliang, et al. "Disentangling Structured Components: Towards Adaptive, Interpretable and Scalable Time Series Forecasting." IEEE Transactions on Knowledge and Data Engineering (2024).
>
> ---
> **Q2:** Thank you for highlighting this point. The decision to exclude an attention mechanism from the long-term component was made after careful consideration. Although it can be incorporated, as we have done with the seasonal, short-term, and spatial components, we found that it brought little to no gain in forecasting accuracy across the datasets we analyzed. To clarify, the attention mechanism is beneficial when a component exhibits a significantly evolving distribution over time. It reduces estimation bias by assigning higher weights to data points with higher correlation. However, for the long-term component in our case, the distribution tends to be stable throughout the input period. In such scenarios, the attention mechanism does not contribute additional value, rendering it redundant and unnecessary. We will include this explanation in the main text of the final version to enhance the coherence of our methodology and improve reader comprehension.
>
> ---
> **Q3:** The 4x4 blocks in Figure 1 are used to exemplify the selection maps $\mathcal{I}^*$ and $\mathcal{E}^*$ as defined in the main text, with $T_{in}$  and $T_{out}$  both instantiated as 4. To enhance clarity, we will explicitly label these blocks in the figure to indicate that they are examples of the associated selection maps. Additionally, we will address the inconsistent text formatting, such as the use of E, to ensure consistency between the figure and the main text.

---

> > ### Comment · Reviewer_aFLy · 2024-08-11
> >
> > I appreciate that you have addressed most of my concerns. I have one additional suggestion: the answer to Q2 could be further strengthened by including some empirical results to support your decision.

---

> > > ### Author Response · Authors · 2024-08-12
> > >
> > > Thank you for your suggestion. We have conducted additional experiments to provide empirical results supporting our decision. For the Electricity dataset, the accuracy with and without long-term attention was 0.128 and 0.129, respectively. For the Traffic dataset, the results were 0.356 and 0.360, respectively. These findings indicate that the inclusion of long-term attention has a minimal and sometimes negative impact on performance.

---

> > > > ### Comment · Reviewer_aFLy · 2024-08-13
> > > >
> > > > Thanks to the author for the further explanation, my concerns are all addressed.

---

> > > > > ### Author Response · Authors · 2024-08-13
> > > > >
> > > > > Dear Reviewer aFLy,
> > > > >
> > > > > we sincerely value your feedback and the constructive suggestions you've provided for enhancing our paper.
> > > > >
> > > > > Authors

---

### Official Review · Reviewer_oTnm · 2024-07-10

**Soundness:** 3
**Presentation:** 4
**Contribution:** 3
**Rating:** 7
**Confidence:** 4

**Summary:**

This study unveils a groundbreaking approach to time series forecasting, notable for its minimal parameter count. It stands as the first model to consistently outperform state-of-the-art (SOTA) techniques while remaining compact. Unlike prevalent methods such as PatchTST and iTransformer, which are powerful but cumbersome, and emerging methods like TimeMixer and SCNN, which are lightweight yet inadequate for complex tasks, this model achieves superior performance without the associated heft.

**Strengths:**

1. The model consistently delivers superior accuracy compared to state-of-the-art (SOTA) methods while maintaining a minimal model size. This accomplishment distinguishes it from other methods.

2. The framework unifies the ability to capture various patterns in time series data, offering a streamlined and enhanced alternative to existing models built with MLPs or Transformers.

3. The authors conduct extensive experiments, showcasing the model's strong performance compared to selected SOTA models, which are sufficiently representative of the latest advancements in the field.

**Weaknesses:**

1. There is a gap between the introduction and Section 3 regarding the decomposition of the time series into four components. The authors do not explain why these four components are sufficient. For longer sequences, is there a need for more components? Are there references that support this approach? This discussion should be included at the beginning of Section 3.

2. Manually disabling the spatial component for certain datasets appears suboptimal. It would be more effective if the algorithm could automatically determine whether including the spatial component is beneficial for each dataset.

3. The paper's formatting needs improvements. It seems the authors may have additional content to include. Although the figures in the methodology section are clear and informative, resizing and rearranging them could provide more space for adding valuable content to the main text.

**Questions:**

If the authors can address the weak points, I would reconsider the score.

**Limitations:**

The authors have raised the limitation of the model concerning computational efficiency, along with potential solutions.

---

> ### Author Rebuttal · Authors · 2024-08-05
>
> Thank you for your thoughtful review and constructive comments.
>
> ---
> **W1:** Thank you for highlighting this important point. We apologize for not providing sufficient context on the four components used in our model. Disentangling these components has been shown to be effective for time series forecasting in numerous studies [1] [2] [3] [4]. Our approach builds on these established methods with a novel selection mechanism, which provably reduces estimation bias for the considered components. Our method advances the performance of decomposition-based approaches to the SOTA level, which can be achieved by only large-scale models before our work. The fact that our model achieves this with only four components suggests that, regardless of model complexity, the representation power of these large-scale models does not exceed the scope covered by these four components. As you pointed out, this raises a critical question for future research: are there additional components that could be modeled to further improve forecasting accuracy, especially for longer sequences? Currently, this question remains open and unanswered. We will discuss this issue and provide relevant references at the beginning of Section 3 in the revised version, ensuring a more comprehensive and informative explanation.
>
> [1] Cleveland, Robert B., et al. "STL: A seasonal-trend decomposition." J. off. Stat 6.1 (1990): 3-73.
>
> [2] Wen, Qingsong, et al. "RobustSTL: A robust seasonal-trend decomposition algorithm for long time series." Proceedings of the AAAI conference on artificial intelligence. Vol. 33. No. 01. 2019.
>
> [3] Deng, Jinliang, et al. "Disentangling Structured Components: Towards Adaptive, Interpretable and Scalable Time Series Forecasting." IEEE Transactions on Knowledge and Data Engineering (2024).
>
> [4] Wang, Shiyu, et al. "TimeMixer: Decomposable Multiscale Mixing for Time Series Forecasting." The Twelfth International Conference on Learning Representations (2024).
>
> ---
> **W2:**  Thank you for your insightful suggestion. Our primary aim is to demonstrate the pioneering predictive capability of a purely decomposition-based model with an exceptionally low parameter count—less than 1% of what large-scale models require. The current architecture leverages the statistical characteristics of the dataset of interest, ensuring that the design is not arbitrary. In particular, our analysis of correlation, autocorrelation, conditional correlation, and conditional autocorrelation—visualized in Figure 7—guided our decisions regarding the architecture's configuration and critical hyper-parameters, such as cycle length and the connection between the variables. While these decisions are based on our judgment, they can be automated through a Python script, a straightforward implementation. Despite its simplicity, this method has already resulted in a sensible architecture that outperforms baselines, thereby validating our contributions. We acknowledge that the current method may not achieve the absolute upper limit of performance possible with all potential decomposition-based model configurations. This limitation is discussed in the "Conclusions, Limitations, and Social Impacts" section. We agree that learning-based methods, such as neural architecture search (NAS), offer a promising avenue for further improvement and optimization. This remains an exciting area for future work.
>
> ---
> **W3:**  Thank you for your valuable suggestion. We will enhance the layout and formatting in the final version of the paper. This will involve resizing and rearranging the figures in the methodology section to ensure they are clear and informative while providing additional space for valuable content in the main text. Our goal is to make the main body of the paper more self-contained and comprehensive.

---

> > ### Comment · Reviewer_oTnm · 2024-08-13
> >
> > Thanks for your reply which has addressed all my questions. I have improved my rating.

---

> > > ### Author Response · Authors · 2024-08-13
> > >
> > > Dear Reviewer oTnm,
> > >
> > > we sincerely value your feedback and the constructive suggestions you've provided for enhancing our paper.
> > >
> > > Authors

---

### Official Review · Reviewer_UAfk · 2024-07-28

**Soundness:** 4
**Presentation:** 3
**Contribution:** 3
**Rating:** 7
**Confidence:** 4

**Summary:**

Title: Parsimony or Capability? Decomposition delivers both in long term time series forecasting.

Long term time series forecasting has been an important research problem which applies to different problem domains. This paper proposes a decomposition method which shows significant performance on the benchmarks with less parameters. This method been evaluated extensively on the various datasets and been competitive to existing models. With such approach models can be enhanced to adapt domain characteristics more effectively in various time series applications.

**Strengths:**

1. SSCNN reduces the parameter count substantially compared to traditional models, holding onto less than 1% of the parameters while still performing well across different datasets.
2. The model captures complex data patterns effectively using fewer parameters, utilizing a structured component-based approach with a selection mechanism to improve prediction accuracy.
3. SSCNN excels in time series forecasting, managing diverse data types and confirming its effectiveness through thorough experimentation.
4. SSCNN improves plain feature decomposition by incorporating a selection mechanism. This allows the model to identify fine-grained dependencies at each time step, which is essential for enhancing the accuracy of the decomposed structured components and, consequently, the overall prediction accuracy.
5. Extensive analysis has been performed to validate the method on existing benchmarks and compared with state-of-the-art methods.
6. Supplementary materials are satisfactory and provide explanation about the dataset and the implementation.

**Weaknesses:**

1. Figures lack captions.
2. Include some limitations of the model as well.
3. second contribution and third one looks quite similar.

**Questions:**

1. Please rewrite the contributions to make them clearer (2nd and 3rd).
2. Add descriptions in figure 2 and 3
3. In section D) Implications of Decomposition on Spatial-Temporal Correlations, please correct the captions of figures in temporal and spatial recovery.

**Limitations:**

Authors have adequately addressed the limitations related to computational efficiency and capability of model.

---

> ### Author Rebuttal · Authors · 2024-08-05
>
> Thank you for your thoughtful review and constructive comments.
>
> ---
> **W1:** We apologize for the lack of captions for Figure 2 and Figure 3. We also include the caption for Figure 1, as requested by Reviewer aS9R. The captions for these figures are shown below
>
> Figure 2: Examination of parameter scale and computation scale against the forward window size and the backward window size on the ECL dataset. (a) (c) For short forward window size or backward window size, SSCNN consumes more parameters than DLinear only. As the window size grows, the SSCNN scales up at a much slower inflation rate than DLinear. (b) (d) The consumption of FLOPS by SSCNN ranks in the middle place among SOTAs, less efficient than iTransformer and Autoformer yet more efficient than PatchTST and Crossformer.
>
> Figure 3: Impacts of backward window size. While the performance of other SOTA forecasters does not necessarily benefit from the increased lookback length, SSCNN improved performance on the enlarged lookback window.
>
> Figure1: (a) Embedding: Raw data points are independently mapped to a high-dimensional representation space using a shared 1x1 convolution operator across all variables and horizons. (b) In-sample Inference: The historical time series representations are processed by three temporal attention-based normalization (T-AttnNorm) layers with different instantiations of the attention map ($\mathcal{I}$), followed by a spatial attention-based normalization (S-AttnNorm) layer. This process yields estimations of four structured components ($\mu$) along with the residual ($R$). (c) Out-of-sample extrapolation: The decomposed components corresponding to each variable are individually extrapolated to future horizons using the matrix multiplication (MatMul) operator, with distinct instantiations of the attention map ($\mathcal{E}$). (d) Component fusion: Finally, all four components, combined with the residuals, are input into a polynomial regression layer.
>
>  ---
> **W2:** We apologize for not highlighting the limitations of the model prominently enough. Due to space constraints, we discussed the limitations in the last section together with broader impacts and conclusions. We will discuss the limitations in detail and put them into a new separate section in the revision
>
> Briefly, the identified limitations include: (a) While our approach effectively reduces model size, its computational complexity remains comparable to Transformer-based models, which is more demanding than linear models. This suggests potential for further optimization in terms of computational efficiency. (b) Currently, the combination of the proposed decomposition modules is determined with statistical insight into the data, which may not be optimal. Future work could explore methods for efficiently searching for the optimal combination of these modules. These points outline practical areas for improvement and future exploration. We will ensure these limitations are clearly presented in the final version of the paper.
>
> ---
> **W3:** We apologize for the confusion between the second and third contributions. The second contribution relates to Section 4, where we theoretically explore the equivalence in capability between the patching operator and the decomposition operator. We also highlight the advantage of decomposition in terms of parameter efficiency, suggesting that decomposition can serve as a parameter-efficient alternative to patching, which is often considered essential in many state-of-the-art (SOTA) models. The third contribution pertains to the empirical evaluation of SSCNN against SOTA baselines, detailed in Section 5.
>
> To clarify, we will revise the second contribution as follows:
>
> "We conduct an in-depth theoretical comparison between decomposition and patching, analyzing both capability and parsimony. This analysis challenges the necessity of patching as an essential component in modern time series forecasting models, proposing decomposition as a more parameter-efficient alternative."
>
> ---
> **Q1:** Please check the response to W3.
>
> ---
> **Q2:** Please check the response to W1.
>
> ---
> **Q3:** We will rectify this issue in the final version.

---

> > ### Comment · Reviewer_UAfk · 2024-08-13
> >
> > Thank you so much for addressing my comments.

---

> > > ### Author Response · Authors · 2024-08-13
> > >
> > > Dear Reviewer UAfk,
> > >
> > > we sincerely value your feedback and the constructive suggestions you've provided for enhancing our paper.
> > >
> > > Authors

---

### Official Review · Reviewer_a5mv · 2024-07-29

**Soundness:** 2
**Presentation:** 2
**Contribution:** 3
**Rating:** 6
**Confidence:** 2

**Summary:**

The paper approaches the problem of long term time series forecasting (LTSF) using a compositional technique to reduce the model size without compromising the quality of solution. The proposed technique is a transformer based architecture with a lower number of parameters, and delivers similar performance as state of the art models for LTSF.

The limitation of existing approaches, such as data patching, is that they fail to take into account the spatio-temporal dependencies, and end up with a blow up in the number of latent variables. This results only in a very small improvement even if the model size is increased substantially. The proposed technique in the paper is based on a inference step and an extrapolation step without any information loss.

The paper evaluates the proposed approach, called SSCNN, with seven datasets, which has a combination of regular and volatile patterns. The baseline and state of the art approaches compared against include iTransformer, TimeMixer, and PatchTST. SSCNN consistently achieves the best scores, with respect to MSE and MAE. The paper also conducts ablation studies to show that each new component in the architecture is vital to the performance.

**Strengths:**

The work studies an important and hard problem in time series forecasting which is the problem of efficient and accurate long term forecasting. Compositional techniques have been successful in other areas of AI including reinforcement learning, planning, and finite state controller synthesis. So, it makes sense to apply similar ideas in the space of long term time series forecasting.

**Weaknesses:**

While the high level message is presented well, I found the details of the proposed method and experiments are hard to follow. A running example with the explanation of the new layers will be useful.

The main contribution with respect to results is somewhat hard to grasp and align with the theoretical claims of the paper. Overall, I think there is room for improvement in the presentation of experimental results. I found some missing details in the experimental section that include:

1. Why is SSCNN missing Figure 3(a)?
2. What is the value of T_{out} in Figure 2?
3. What is the forward window size in Figure 3?

In figure 2, it would be useful to move some of the methods to the appendix, and keep only the critical ones  in the main body of the paper. Same is true for Figure 3. It is hard to go back and forth between figures 1, and figures2&3.

Minor:

1. I would suggest providing some more details about the experimental results in point 3 of the contributions (lines 80-82)
2. Figure 3 is hard to read in print.
3. Having only one legend for all the subplots (Figure 2(a)-(d) and Figure 3(a)-(d)) will better than repeating the legends in all subplots.

**Questions:**

1. Are there any non neural network based time series forecasting models which make use of compositionality?
2. Do any of the introduced layers (temporal, seasonal, short-term, etc) have similarities with any existing literature? What I mean is that,  is the novelty in getting the layers to work together, or, also in defining the individual layers?
3. In order to better study the computational cost, can you share the total/average running time for each method for each dataset?
4. Why is SSCNN missing Figure 3(a)?
5. What is the value of T_{out} in Figure 2?
6. What is the forward window size in Figure 3?

**Limitations:**

Yes, the paper has outlined the limitation of computational efficiency and provided some insight into how it can be improved in future work.

---

> ### Author Rebuttal · Authors · 2024-08-05
>
> Thank you for your thoughtful review and constructive comments.
>
> ---
> **W1:** We apologize if the current symbolic formulas present challenges to comprehension. To enhance understanding, we have tried to make the selection mechanisms formulated by Equations (3)-(7) more accessible with the example attention maps provided in Figure 1. To further illustrate the computational flow, we present a toy example focusing on the inference of long-term, seasonal, and short-term components, with the scaling operation and the selection mechanism being disabled for simplicity:
>
> Given a historical sequence of observations: 1.1, 3.1, 1.0, 3.0, 0.9, 2.9:
>
> Long-term Component: Derived as 2, 2, 2, 2, 2, 2, resulting in the sequence of residuals: -0.9, 1.1, -1, 1, -1.1, 0.9.
>
> Seasonal Component: Computed as -1, 1, -1, 1, -1, 1, leading to the residual series: 0.1, 0.1, 0, 0, -0.1, -0.1.
>
> Short-term Component: Derived as 0, 0.1, 0.05, 0, -0.05, -0.1 with a window size of 2, obtaining the residual series: 0.1, 0, -0.05, 0, -0.05, 0.
>
> ---
> **W2:** We apologize for any confusion caused by the perceived disconnect between the experimental results and the theoretical claims of the paper. The experimental results presented in Table 1 demonstrate SSCNN's improved efficacy compared to state-of-the-art methods, including the patching-based methods PatchTST and iTransformer. These results align with the theoretical claims made in Section 4.1, suggesting that SSCNN's decomposition-based approach either matches or exceeds the capabilities of patching-based methods. Additionally, Figure 2 illustrates the optimal parsimony of SSCNN, supporting the analysis in Section 4.2. Furthermore, Figure 3 highlights SSCNN's superior scalability with an increasing lookback window size. Unlike other baselines, SSCNN consistently benefits from more historical data points, regardless of the dataset's dynamic nature. This scalability is crucial for time series forecasting, as more data generally leads to more accurate predictions following our expectation.
>
>  ---
> **W3:**  (1). Thank you for catching this error. The label "HDformer" in Figure 3(a) is indeed a typo and should be replaced with "SSCNN." We will correct this in the revised version.
>
> (2). In Figure 2, for subfigures (a) and (b), $T_{in}$  is set to 96, while $T_{out}$  is represented by the x-axis. For subfigures (c) and (d), $T_{out}$  is set to 96, while $T_{in}$  is represented by the x-axis. We will clarify this information in the revised figure caption to improve understanding.
>
> (3). The forward window size in Figure 3 is indicated by $T_{out}$, which is noted in each subfigure. We will make sure this information is more clearly presented in the revised version for better clarity.
>
> ---
> **W4:** We will revise the third contribution as follows in the final version:
> We conducted comprehensive experiments to evaluate SSCNN across various dimensions. SSCNN demonstrates improved effectiveness, with increases ranging from 4% to 20%, and a reduction in model size ranging from 70% to 99% on average across all datasets, compared to state-of-the-art baselines.
>
> ---
> **W5:** We will also address the other concerns you raise about the layout and formating in the final version.
>
> ---
> **Q1:** The concept of compositionality in time series forecasting has roots in classical statistical methods, most notably STL [1]. Our method builds on this foundational idea by integrating it with deep learning techniques, enabling nonlinear decomposition and re-composition of the time series data. Additionally, we also extend STL to consider the short-term and spatial components. In this way, our method achieves a more nuanced and effective modeling of complex time series patterns.
>
> [1] Cleveland, Robert B., et al. "STL: A seasonal-trend decomposition." J. off. Stat 6.1 (1990): 3-73.
>
> ---
> **Q2:** The novelty of our approach lies in the reinvention of these layers that are inspired from the existing literature. As mentioned in the introduction and related work sections, some existing works have also explored the idea of modeling different heterogeneous components separately. These works typically employed techniques such as plain moving averages or MLPs. However, they often struggled to outperform advanced baselines like PatchTST across various benchmark datasets, as shown in Tables 1 and 2. Moreover, these approaches lacked scalability, requiring exponentially more parameters than SSCNN for large forward and backward window sizes, as demonstrated in Figures 2(a) and 2(c). Our method significantly advances these prior approaches by integrating customized selection mechanisms for the components of different behaviors within both backward and forward estimation processes. This extension enhances our model's capability and parsimony, resulting in superior performance.
>
> ---
> **Q3:** We have preliminarily reported the measurements of training and inference times for SSCNN compared to two representative baselines, PatchTST and iTransformer, on the Electricity and ETTh1 datasets. We will improve this evaluation study in the revision. Our preliminary findings indicate that iTransformer is the most efficient model among the three, requiring the least time for both training and inference. The relatively higher running time of SSCNN can be in part attributed to its sequential handling of the considered components. In contrast, iTransformer handles these components in parallel using a unified MLP, contributing to its efficiency advantage. We have mentioned this limitation of SSCNN in the section of "Conclusions, Limitations and Broader Impacts".
>
> | Models| ECL |  ETTh1 |
> |-|-|-|
> |iTransformer | 11min (training), 0.05s (inference) |  2min (training), 0.01s (inference) |
> PatchTST|95 min (training), 0.3s (inference)|10min (training), 0.09s (inference)
> |SSCNN|72 min (training), 0.2s(inference)|8min (training), 0.07s (inference) |
>
> ---
> **Q4, Q5, Q6:** Please check the above response to W3.

---

> > ### Comment · Reviewer_a5mv · 2024-08-12
> > **Thanks for responding to my queries**
> >
> > I would like to thank the authors for addressing my questions thoroughly.
> > I have more clarity on the novelty aspect now.

---

> > > ### Author Response · Authors · 2024-08-13
> > >
> > > Dear Reviewer a5mv,
> > >
> > > we sincerely value your feedback and the constructive suggestions you've provided for enhancing our paper. If you have any further questions or concerns, please feel free to let us know.
> > >
> > > Authors

---

### Author Rebuttal · Authors · 2024-08-05

We are grateful for the detailed and constructive feedback provided by the reviewers. The positive reception of our work, particularly the recognition of its innovative approach and significant contributions to the field of time series forecasting, is highly encouraging. Below, we summarize the key strengths of our paper as highlighted by the reviewers:

1. **Importance and Relevance** (**Reviewer a5mv**): Acknowledged the significance of tackling efficient and accurate long-term time series forecasting.

2. **Innovative Approach and Original Contributions** (**Reviewers a5mv, UAfk, p9WT, oTnm, aS9R**): Highlighted the innovative use of decomposition with a selection mechanism. The reviewers appreciated the originality of the approach in feature decomposition, effectively capturing complex data patterns with a minimal model size. The work addresses a critical gap in SOTA methods, demonstrating the potential for smaller models to match or surpass existing methods.

3. **Lightweight and Strong Performance** (**Reviewers UAfk, oTnm, aFLy, p9WT**): Noted the substantial reduction in parameter count while maintaining or surpassing the performance of SOTA models. The reviewers appreciated the model's capability to handle diverse datasets and achieve state-of-the-art accuracy.

4. **Comprehensive Evaluation** (**Reviewers UAfk, aS9R, aFLy, p9WT**): Praised the extensive and thorough experiments, including overall performance results, ablation studies, and comparisons with other SOTA methods. The reviewers also noted the consistent performance across different experimental conditions and the clear documentation of results.

5. **Clarity and Quality** (**Reviewers aS9R, UAfk, p9WT**): Commended the clarity of the writing, logical flow, and the comprehensibility of the paper. Reviewers appreciated the supplementary materials and clear usage instructions for the open-source code, which facilitated reproducibility.

We have carefully addressed each of the reviewers' concerns and have made corresponding revisions and clarifications throughout the manuscript. We believe these changes enhance the clarity, coherence, and overall quality of the paper. We invite the reviewers to refer to the specific reply box associated with their review for detailed responses and updates. Thank you for your thoughtful consideration, and we look forward to your feedback on the responses.

---

### Comment · Area_Chair_XFv7 · 2024-08-12
**Thank you, Authors!**

Dear Authors,

Thank you for your responses to Rviewers and your categorized, succinct overall response. On our end, we will collectively continue to review these.

I hope the Reviewers' reviewers and discussion have helped strengthen your work and highlighted its significance and areas for improvement.

Thank you for your submission and continued effort in the review process!

Best,

Area Chair xFv7

---

### Decision · Program_Chairs · 2024-09-25

**Decision:**

Accept (spotlight)

**Comment:**

The paper proposes a novel decomposition-based approach for long-term time-series forecasting, which significantly reduces model complexity while maintaining or surpassing the performance of state-of-the-art methods. The model is particularly effective in capturing complex data patterns with minimal parameters, offering a robust and scalable solution for diverse datasets.

**Strengths**
- Parameter Efficiency: The model achieves superior performance using over 99% fewer parameters than competing methods, addressing a critical challenge in long-term time-series forecasting problems that generally require more parameters to maintain history and to avoid catastrophic forgetting.
- Enhanced Interpretability: The decomposition approach not only improves accuracy but also enhances model interpretability, making the results more actionable in practical applications, supporting confidence in its value in real-world scenarios (Reviewers UAfk and aFLy).
- Comprehensive Empirical Validation: The paper presents extensive empirical evaluations across multiple datasets, showcasing the generalizability and effectiveness of the proposed method. Reviewer UAfk specifically praised the thoroughness of the experiments and the clear demonstration of the model’s superiority over existing SotA methods.
- Clarity and Reproducibility: The writing is generally clear, and the code is well-documented and reproducible, facilitating future research and application.

**Areas Needing (1st, 2nd) or Recommended for (3rd) Improvement (Weaknesses)**
- Methodological Clarity: Aspects of the methodological decisions, particularly the choice of decomposition techniques, lack clarity (Reviewer aFLy).
--Providing more detailed justifications and discussions on these choices would improve the paper’s overall robustness in the claims of their algorithm's capabilities and results.
- Presentation and Formatting Issues: (Reviewer aS9R) highlighted minor presentation issues that detract from the paper’s readability. While these do not undermine the scientific contribution, they should be addressed to ensure the paper’s professionalism and clarity.
- Limited Comparison with LLM-based Models: The paper would benefit from a more detailed comparison with large language model (LLM)-based approaches, which are increasingly relevant in the forecasting domain. Addressing this would strengthen the paper’s position within the broader research landscape. (Reviewer p9WT)

The paper offers a significant contribution to long-term time-series forecasting through its innovative approach, sound experimental design, and comprehensive empirical validation.

The identified issues 1 and 2 are important and easily addressable. *If* this paper is accepted, these issues must be corrected for the camera-ready version. Improve the methodological clarity and aim to improve presentation and formatting issues. Additionally, comparisons with LLM-based models are suggested for future work but are not required nor expected for the current work.

---

**[Update: 9-20-24]** After manually reviewing the paper in entirety as well as all reviews, rebuttals, and discussions a week ago, I believe that this work shows progress in multiple key areas time-series forecasting, which is a large, growing domain that will continue to grow over time. Namely this progress is demonstrated in (1) better forecasting performance over multiple time horizons on (2) several, diverse problem settings against (3) state-of-the-art and/or reasonably comparable baselines, with (4) order of magnitudes fewer parameters than comparable models due to a focus on decomposition.

The Authors proactively list limitations in the time taken for training and inference versus certain alternative approaches and provide sound logical reasoning as to why this is the case. This along with the inclusion of other limitations helps the research community in identifying where they should or should not focus their efforts.

As a result of the strong generalizable results in a large problem domain due to an innovative approach that deviates from the norm in a refreshing way, I recommend this paper for acceptance and spotlight.